# Accurately Segmenting/Mapping Tobacco Seedlings Using UAV RGB Images Collected from Different Geomorphic Zones and Different Semantic Segmentation Models

**DOI:** 10.3390/plants13223186

**Published:** 2024-11-13

**Authors:** Qianxia Li, Zhongfa Zhou, Yuzhu Qian, Lihui Yan, Denghong Huang, Yue Yang, Yining Luo

**Affiliations:** 1School of Geography & Environmental Sciences, Guizhou Normal University, Guiyang 500025, China; lqx2384730355@163.com (Q.L.); 242100170587@gznu.edu.cn (Y.Q.);; 2School of Karst Science, Guizhou Normal University, Guiyang 550003, China; 3State Engineering Technology Institute for Karst Desertification Control, Guiyang 550000, China

**Keywords:** tobacco seedling plants, UAV RGB imagery, semantic segmentation model, different geomorphic zones

## Abstract

The tobacco seedling stage is a crucial period for tobacco cultivation. Accurately extracting tobacco seedlings from satellite images can effectively assist farmers in replanting, precise fertilization, and subsequent yield estimation. However, in complex Karst mountainous areas, it is extremely challenging to accurately segment tobacco plants due to a variety of factors, such as the topography, the planting environment, and difficulties in obtaining high-resolution image data. Therefore, this study explores an accurate segmentation model for detecting tobacco seedlings from UAV RGB images across various geomorphic partitions, including dam and hilly areas. It explores a family of tobacco plant seedling segmentation networks, namely, U-Net, U-Net++, Linknet, PSPNet, MAnet, FPN, PAN, and DeepLabV3+, using the Hill Seedling Tobacco Dataset (HSTD), the Dam Area Seedling Tobacco Dataset (DASTD), and the Hilly Dam Area Seedling Tobacco Dataset (H-DASTD) for model training. To validate the performance of the semantic segmentation models for crop segmentation in the complex cropping environments of Karst mountainous areas, this study compares and analyzes the predicted results with the manually labeled true values. The results show that: (1) the accuracy of the models in segmenting tobacco seedling plants in the dam area is much higher than that in the hilly area, with the mean values of mIoU, PA, Precision, Recall, and the Kappa Coefficient reaching 87%, 97%, 91%, 85%, and 0.81 in the dam area and 81%, 97%, 72%, 73%, and 0.73 in the hilly area, respectively; (2) The segmentation accuracies of the models differ significantly across different geomorphological zones; the U-Net segmentation results are optimal for the dam area, with higher values of mIoU (93.83%), PA (98.83%), Precision (93.27%), Recall (96.24%), and the Kappa Coefficient (0.9440) than those of the other models; in the hilly area, the U-Net++ segmentation performance is better than that of the other models, with mIoU and PA of 84.17% and 98.56%, respectively; (3) The diversity of tobacco seedling samples affects the model segmentation accuracy, as shown by the Kappa Coefficient, with H-DASTD (0.901) > DASTD (0.885) > HSTD (0.726); (4) With regard to the factors affecting missed segregation, although the factors affecting the dam area and the hilly area are different, the main factors are small tobacco plants (STPs) and weeds for both areas. This study shows that the accurate segmentation of tobacco plant seedlings in dam and hilly areas based on UAV RGB images and semantic segmentation models can be achieved, thereby providing new ideas and technical support for accurate crop segmentation in Karst mountainous areas.

## 1. Introduction

Tobacco (*Nicotiana tabacum* L.) is a plant of the Solanaceae family that is native to South America. As an important economic crop, it is planted in over 125 countries worldwide, with a global estimated planting area of over 4 million hectares. China is the largest country in tobacco cultivation, with areas in over 1700 counties and cities in 27 provinces being used for this purpose. In 2021, the overall area for tobacco cultivation in China reached nearly 1 million hectares. The economic value of tobacco not only contributes greatly to national and local financial income and economic development, but tobacco also has a variety of significant medical properties [1,2]. Guizhou has a long history of tobacco cultivation due to its unique ecological environment, subtropical monsoon warm humid climate that is most suitable for tobacco growth, abundant rainfall and light and heat resources, and advantageous natural conditions in terms of climate, soil, topography, and geomorphology; therefore, it is an ideal production area for high-quality roasted cigarettes and is the second largest production area in the country [3,4].

Guizhou is located in Southwestern China in the central hinterland of one of the world’s three major Karst regions. The Karst region has an area of 109,084 km^2^, accounting for 61.9% of the province’s total land area, with mountains and hills accounting for 92.5% of the area. Guizhou is the only province in the country that does not have plain dams and wide river valleys. Because of its undulating terrain, fragmented surface, vertical and horizontal valleys, and soil-forming factors, there are many large sloping farmland areas with a wide and scattered distribution within the territory. At the same time, the Karst mountainous areas have more people and less flat land. Tobacco is mostly planted on sloping farmland with a complex terrain and fragmented plots. The distribution of planting space is scattered, the background characteristics of planting plots are complex, and there are numerous cases of crop intercropping. Furthermore, the tobacco cultivation process carries high risks, as it is susceptible to natural disasters and pests. Therefore, timely information on the spatial distribution of the tobacco planting area, growth, yield, and disaster loss is important to achieve refined tobacco management and accurate yield estimation, which can assist in governmental decision-making [5,6,7,8]. Yield estimation plays an important role in tobacco planting management and agricultural precision [9,10], as fast and accurate extraction of tobacco seedling plants is an important prerequisite for precise fertilization, timely replanting, and yield estimation.

Precision segmentation of tobacco seedlings is a crucial step in seedling replanting, precision fertilizer application, and subsequent yield estimation. Freshly transplanted tobacco seedlings are susceptible to natural disasters such as low temperatures, flooding, high winds, and hailstorms, which cause varying degrees of losses to tobacco production, while residual mulch and field weeds affect the growth of tobacco and reduce its quality and yield [11]. During the tobacco seedling stage, the growth of transplanted tobacco seedlings can be checked in a timely manner. Timely replanting should be carried out for dead, weak, or missing seedlings. It is advisable to water withered tobacco seedlings in a timely manner to facilitate their normal growth [12,13]. Therefore, accurate monitoring of tobacco seedlings can solve the problem of seedling replanting in a timely manner, as well as help to strengthen the management of tobacco plants and weeds in the field, disease and pest prevention, and other measures, to ensure the optimal growth of tobacco plants and improve the quality. This, in turn, improves the economic and medicinal value of tobacco. In addition, the quantification of tobacco seedlings in plots is an important prerequisite for accurate yield estimation in precision agriculture. However, traditional tobacco seedling counting relies on manual counting, which is time-consuming and labor-intensive, and the counting results are susceptible to subjective bias, making it more challenging to monitor tobacco growth in complex scenarios. Therefore, further research on tobacco seedling counting techniques is particularly important.

Previous studies have focused on the application of remote sensing technology in tobacco acreage extraction [14,15]. Huang et al. [9] trained four deep semantic segmentation models—namely, DeeplabV3+, PSPNet, SegNet, and U-Net—to extract tobacco plantation areas based on UAV images. Zhu X et al. [16] combined supervised classification with image morphological operations to carry out the extraction of tobacco field areas. Zhang Yang et al. [17] extracted and analyzed the spectral characteristics of the features and vegetation indices using Sentinel-2A images, and used a decision tree classification method to extract the tobacco planting area. As the fine extraction of tobacco seedlings requires remote sensing images with a high spatial resolution, the present study mainly focuses on the extraction of information from tobacco planting areas. There are limited studies on the precise extraction of single tobacco plants (especially the extraction of tobacco seedlings) directly from images using remote sensing technology. It is even more challenging to extract data on tobacco plant seedlings using satellite images.

In recent years, UAV remote sensing technology has been rapidly developed and has become widely used in various fields. The main reason is the ability of drones to be equipped with sensors that capture images across multispectral, hyperspectral, and visible light, while being unrestricted by spatial, temporal, and spectral resolutions, as well as complex weather. Therefore, high-spatial-resolution images of complex terrain in Karst mountainous areas can be obtained in a flexible, economical, and efficient manner, compensating for the shortcoming of not being able to obtain real-time high-resolution satellite images. As UAV remote sensing provides spatial distribution information on crops, it has been widely used in precision agriculture [18,19,20,21,22,23]. For example, Xia Yan et al. [24] proposed a UAV image tobacco fine extraction method combining multi-feature and ultra-pixel segmentation, achieving an accuracy of 99% for the difficult problem of extracting tobacco monocultures. Honghui Xie [25] used watershed and center-region segmentation to select tobacco candidate regions, selected multiple features, and used a support vector machine to extract tobacco plants. Xie H et al. [26] converted tobacco images into lab color space, performed the morphological reconstruction of B-channels, extracted tobacco candidate regions based on the processed B-channels, and selected multiple features to extract tobacco plants using a support vector machine. Fan Z et al. [27] extracted multiple candidate regions of tobacco plants via morphological manipulation and watershed segmentation based on the property that the center regions of tobacco plants are brighter than the leaf regions, constructed a deep convolutional neural network to extract tobacco and non-tobacco plants, and later performed non-tobacco plant removal. Chen Daoying et al. [28] classified images of a tobacco field using K-means clustering to obtain an initial result containing both tobacco plants and weeds, extracted multiple features, and used a BP neural network to reject weeds and extract tobacco plants, achieving an average accuracy of more than 90%. The fine extraction of tobacco is mainly accomplished through the combination of super-pixel segmentation and supervised classification, but the above methods still have many limitations, such as low extraction accuracy and low automation.

With the improvement in computer hardware, software, and arithmetic power, deep learning has attracted great attention in scientific research and practical applications, and many scholars have applied it in the field of agriculture. The combination of UAV remote sensing technology and deep learning has led to the rapid development in precision agriculture. In recent years, deep learning methods for RGB imaging crop counting have emerged [29,30]. Samii et al. [31] designed and trained a deep learning CNN network to learn about cotyledon opening during plant seedling development. Jiang et al. [32] used the Faster-RCNN model together with the Inception ResNetv2 feature extractor to accurately compute plant seedlings in the field. Yang et al. [33] introduced Yolov4-based spatial pyramid pooling (SPP) and multilevel feature fusion methods, achieving substantial improvements in the counting performance. These methods use the detected object boxes for counting, and the output is the position of each instance and its corresponding bounding box. However, the number of tobacco seedlings in a field can range from a few to thousands, especially in Karst mountainous areas where the land is fragmented and the surface is complex. Tobacco is planted in environments with numerous weeds and mixed crops, making it difficult to accurately segment tobacco seedlings. In addition, the large number of tobacco seedling samples brings an enormous workload to the labeling process.

This study explores a family of tobacco plant segmentation networks—namely, U-Net, U-Net++, Linknet, PSPNet, MAnet, FPN, PAN, and DeepLabV3+—for the accurate segmentation of tobacco seedlings based on semantic segmentation models of UAV-RGB images. The aims of the study are as follows: (I) selection of the most suitable model for the accurate segmentation of tobacco plant seedlings in complex Karst mountains; (II) construction of tobacco seedling datasets from different geomorphologic regions to train different semantic segmentation models; (III) manual evaluation of the performance of different models in different geomorphologic zones; and (IV) field validation of the proposed semantic segmentation model.

## 2. Materials and Methods

### 2.1. Study Area

Guizhou has a long history of tobacco planting; however, due to its terrain, surface fragmentation, climate, and other factors, the arable land resources are very limited. Crop cultivation is diverse to effectively utilize such resources and intercropping is common, resulting in an extremely complex tobacco planting environment. In view of the complex environmental conditions for tobacco cultivation in Guizhou, two major tobacco-growing towns—Beipanjiang Township (25°36′08′′ N, 105°35′53′′ E) in Zhenfeng County and Qiangsiang Township (25°10′48′′ N, 105°11′24′′ E) in Anlong County, Qianxinan Prefecture—were selected as the study areas (Figure 1).

Beipanjiang Township has a Karst hilly landscape, and the northern part of the area comprises more than 95% stone nooks and crannies. The soil and climate conditions are extremely suitable for the growth of peppercorns, sand nuts, and other cash crops, and the area has been developed into a three-dimensional agricultural production area for peppercorn and sand nut planting. The central and southern areas, with a relatively flat terrain, are dominated by the cultivation of high-quality tobacco, greenhouse vegetables, corn, and other crops, forming an agricultural industrial structure with complementary advantages. Qianxiang Township is mainly hilly, with a landscape composed of mountainous basins and grooves. The dam area has a lot of arable land and is an important agricultural industrial base in Anlong County, with the main cash crops being tobacco, rice, corn, and oilseed rape.

### 2.2. Data Acquisition

The DJI Mavic 2 Pro drone (Shenzhen, China) is compact and ultra-lightweight, capable of being equipped with a variety of sensors, including the 1-inch CMOS Hasselblad camera. This camera is ideal for capturing high-quality remote sensing data, such as visible light (RGB) imagery, which can be utilized for monitoring tobacco plants in the study area. The time of image collection was during the seedling stage of tobacco planting, and the drone shooting time in the study area of Beipanjiang Township was 14:00–16:00 on 4 June 2021, under clear and cloudless weather and good visibility conditions, while the shooting time in the study area of Qianxiang Township was 14:00–16:00 on 6 June 2023, under clear and cloudy weather and better visibility conditions. All of these conditions met the requirements for safe drone operation. When performing the flight mission, the heading and side overlap rates were set to 80% to ensure the accuracy and quality of the captured tobacco images, and the waypoint hovering method was used for shooting. RTK (USA) was used to collect the ground control points, and the images collected using the UAV were geographically aligned and spliced using the Pix4D 4.4.12 (Redlands, California, USA) commercial software. Figure 2 shows the UAS (Figure 2A), RTK (Figure 2B), and Pix4D software (Figure 2C).

### 2.3. Construction of the Seedling Tobacco Dataset

Traditionally, sampling, data collection and production, and crop labeling need to be performed in the field to obtain data on the relevant crops, which is time-consuming and labor-intensive. This study used high-resolution UAV images and the ArcMap 10.2 software to label crops, which greatly reduces the working time and cost.

The data volume of the high-resolution UAV remote sensing images in the study areas is large, and their direct use as samples is not conducive to the training of deep learning models due to the limited computer performance and computational capacity; therefore, the ArcMap software was used to manually and accurately annotate along the contours of tobacco plants and binarize the tobacco images to produce tobacco plant labels. The tobacco images were randomly segmented and manually annotated into 224 pixel × 224 pixel samples to obtain an accurately labeled tobacco seedling dataset. To better explore the impact of the datasets on model training performance, this study produced three datasets for deep learning model training based on the degree of terrain undulation and fragmentation of arable land in the Guizhou Karst terrain. The three datasets included a Hill Seedling Tobacco Dataset (HSTD), a Dam Area Seedling Tobacco Dataset (DASTD), and a Hilly Dam Area Seedling Tobacco Dataset (H-DASTD). The process of tobacco seedling dataset construction is shown in Figure 3, and detailed descriptions of the HSTD, DASTD, and H-DASTD are shown in Table 1.

### 2.4. Segmentation Model Selection for Tobacco Plant Seedlings

With the development of deep learning and computational hardware technologies, many new deep learning models have been proposed and used for image segmentation. However, the applicability of each model should be considered. There is an urgent need to select the most suitable deep learning model for the segmentation of tobacco seedlings in Karst mountainous areas, due to their rugged terrain and complex crop planting environments, which lead to difficulties in obtaining high-precision image data. In order to reduce the sample size, operation time, and computational complexity, as well as improve the segmentation accuracy and explore the applicability of different semantic segmentation models for the segmentation of complex underground tobacco seedlings in Karst mountainous areas, eight classical semantic segmentation models were selected as the segmentation models for tobacco seedlings, namely, U-Net, U-Net++, Linknet, PSPNet, MAnet, FPN, PAN, and DeepLabV3+. Table 2 provides a detailed overview of when each model was proposed, its framework, and application.

### 2.5. Training Environment

During the training process, a semantic segmentation model performs numerous calculations and occupies memory and video memory features; thus, the operation of the model has high hardware requirements. In the study framework, we used TensorFlow 2.0.0 to build a DL research environment and test the U-Net, U-Net++, Linknet, PSPNet, MAnet, FPN, PAN, and DeepLabV3+ models. The hardware and software environments are described in Table 3.

### 2.6. Assessment of Indicators

After training the models on tobacco plant segmentation, five evaluation metrics—namely, Mean Intersection over Union (mIoU), Pixel Accuracy (PA), Precision, Recall, and the Kappa Coefficient—were selected to quantify the segmentation results and evaluate the performance of each model. The calculated metrics were compared to determine the most appropriate semantic segmentation model for the segmentation of tobacco seedlings in the complex planting environments of Karst mountainous areas. mIoU is an evaluation index of the model prediction results: the closer the value of mIoU to 1, the better the model’s segmentation effect, and the closer the value to 0, the worse the model’s segmentation effect. PA is used to assess the global accuracy of a model and represents the number of samples that the model correctly classifies as a tobacco growing area divided by the number of total samples. Recall is the ratio of pixels correctly recognized as tobacco plant pixels among all the tobacco planting area pixels in a prediction image, and is used to evaluate the effectiveness of the prediction results. A superior value of these evaluation metrics indicates the high accuracy and robustness of a model in segmenting tobacco seedling plants. The formulae for calculating the metrics are as follows:$$mIoM=\frac{1}{2}\left(\frac{TP}{TP+FP+FN}+\frac{TN}{TN+FN+FP}\right)$$
$$PA=\frac{TP+TN}{TP+TN+FP+FN}$$
$$\mathrm{Pr}ecision=\frac{TP}{TP+FN}$$
Recall=TPTP+FN
$$Kappa=\frac{p_{o}-p_{e}}{1-p_{e}}$$
where *p_o_* denotes the number of correctly recognized phase elements (i.e., the overall classification accuracy) and *p_e_* is the ratio of the sum of the products of the actual and predicted data volumes corresponding to all the categories to the square of the total number of samples, which are two parameters that express the degree of how good or bad the segmentation results are. *TP* is the number of pixels where the true value and the predicted segmentation result both indicate a tobacco plant; *FP* is the number of pixels where the predicted segmentation result indicates a tobacco plant but the true value is not a tobacco plant; *FN* is the number of pixels where the true value is a tobacco plant but the predicted segmentation result indicates it is not a tobacco plant; and *TN* is the number of pixels where neither the true value nor the predicted result indicate a tobacco plant.

## 3. Results and Discussion

Much progress has been made in pixel-level semantic segmentation, driven by powerful deep neural networks. However, it is extremely challenging to perform pixel-level information extraction in complex natural scenes. To this end, this study applied U-Net, U-Net++, Linknet, PSPNet, MAnet, FPN, PAN, and DeepLabV3+ to the segmentation of tobacco seedling plants in the complex planting environments of Karst mountainous areas, and the results were comparatively analyzed to evaluate each model’s performance in tobacco seedling recognition. U-Net, U-Net++, Linknet, PSPNet, MAnet, FPN, PAN, and DeepLabV3+ were first pre-trained using the HSTD, DASTD, and H-DASTD to obtain the most suitable model for tobacco seedling segmentation. Then, mIoU, PA, Precision, Recall, and the Kappa coefficient were used to quantitatively evaluate the model training efficiency and accuracy in segmenting tobacco seedling plants.

### 3.1. Analysis of the Optimal Segmentation Model for Tobacco Plants

This study constructed the HSTD, DASTD, and H-DASTD for semantic segmentation model training, and evaluated and analyzed the accuracy of different semantic segmentation models using five quantitative indices, namely, mIoU, PA, Precision, Recall, and the Kappa Coefficient. The aim was to obtain an optimal model through the training and extraction of tobacco seedling plants from three kinds of datasets collected from hilly, dam, and hilly dam areas under complex planting environments in Karst mountainous areas.

#### 3.1.1. Analysis of the Results of HSTD Training Model Segmentation

The selected semantic segmentation models for segmenting tobacco seedlings in the hilly area were evaluated by comparing mIoU, PA, Precision, Recall, and the Kappa Coefficient. Table 4 shows the segmentation results of the eight semantic segmentation models, with each trained model being evaluated against a test dataset.

As shown in Table 4, the U-Net++ model achieved the best segmentation accuracy, with mIoU and PA values of 84.35% and 98.56%, respectively. MAnet achieved the highest recall of 81.14%, with a value of 83.77% for mIoU, 98.47% for PA, 73.85% for Precision, and 0.7704 for the Kappa Coefficient. Among the eight models, PSPNet had the worst segmentation performance, with mIoU, PA, Precision, Recall, and Kappa Coefficient values of 71.46%, 97.27%, 63.23%, 52.16%, and 0.5670, respectively. Through an analysis of these five indicators, the overall PA value was found to be the best, with all eight models having a PA value over 96%. The overall performance of the Kappa Coefficient was the worst, with a maximum value of only 0.78 achieved by Linknet. Meanwhile, the accuracy of the segmentation results of these models was found to vary greatly, with mIoU improving from 71.46% (PSPNet) to 84.17% (U-Net++), a difference of 12.71%; PA improving from 96.74% (Linknet) to 98.56% (U-Net++), a difference of 1.82%; Precision improving from 63.23% (PSPNet) to 81.55% (Linknet), a difference of 18.32%; Recall improving from 52.16% (PSPNet) to 81.14% (Manet), a difference of 28.98%; and the Kappa Coefficient improving from 0.5670 (PSPNet) to 0.7800 (Linknet), a difference of 0.213. Therefore, the research results indicate that the overall segmentation performances of U-net, U-Net++, Linknet, and PSPNet were better than those of MAnet, FPN, PAN, and DeepLabV3+.

Overall, the eight deep learning models showed a poor segmentation performance for tobacco seedlings in the complex hilly area, with Kappa Coefficients below 0.8. Through comparing and analyzing the performances of the eight models, it can be concluded that both U-Net and U-Net++ had a better segmentation performance, indicating that they are more suitable for the segmentation of tobacco seedlings in hilly areas than the other six models, and that U-Net and U-Net++ can be applied to extract information about crops in complex planting environments. Figure 4 presents the comparison results of the performance of the eight different DL models in segmenting tobacco seedling plants in the hilly area.

#### 3.1.2. Analysis of the Results of the DASTD Training Model Segmentation

Table 5 presents the segmentation results of the eight semantic segmentation models trained on the DASTD. The performance of the selected models in segmenting tobacco seedlings in a dam area was evaluated by comparing the mIoU, PA, Precision, Recall, and Kappa Coefficient values. It was found that the models achieved excellent results in segmenting tobacco seedlings in the dam area, with mIoU, PA, Precision, Recall, and the Kappa Coefficient reaching values of at least 87%, 97%, 91%, 85%, and 0.8080, respectively, Therefore, it can be concluded that, for dam areas with simple planting environments, using deep learning models can achieve precise segmentation of tobacco seedlings.

As shown in Table 5, the U-Net model achieved the best results in segmenting tobacco seedlings in the entire dam area, with mIoU, PA, Precision, Recall, and Kappa Coefficient values of 93.83%, 98.83%, 93.27%, 96.24%, and 0.9440, respectively. Among the eight models, PSPNet had the worst segmentation results, with mIoU, PA, Precision, Recall, and Kappa Coefficient values of 87.78%, 97.66%, 91.18%, 86.44%, and 0.8080, respectively. Meanwhile, it was observed that the difference in the segmentation accuracy of the models was more obvious, with a difference of 6.05% for mIoU, 1.17% for PA, 2.09% for Precision, 10.66% for Recall, and 0.136 for the Kappa Coefficient. Based on the comparison of mIoU, it was found that for the segmentation of tobacco seedlings in the dam area, the higher the mIoU value, the better the model performance; in particular, U-Net (93.83%) > U-Net++ (92.90%) > Linknet (92.78%) > MAnet (92.74%) > FPN (92.57%) > DeepLabV3+ (92.22%) > PAN (90.02%) > PSPNet (87.78%).

Overall, for the segmentation of tobacco seedlings in the dam area, U-Net achieved the best overall performance while PSPNet had the lowest performance. Therefore, U-Net is most suitable for crop segmentation in dam areas with simple planting environments, and it can be applied as an effective model in various crop segmentation tasks. Figure 5 presents the comparison results of the performance of the eight different DL models in segmenting tobacco seedling plants in the dam area.

#### 3.1.3. Analysis of the Results of H-DASTD Training Model Segmentation

Table 6 shows the accuracy of the segmentation results of U-Net, U-Net++, Linknet, PSPNet, MAnet, FPN, PAN, and DeepLabV3+ trained on the H-DASTD, taking into account the effect of the hilly and dam planting environments on the segmentation performance of these deep learning models.

Table 6 shows that U-Net achieved the best performance with the highest accuracy compared with the other deep learning models in segmenting tobacco seedling plants in the hilly dam area. Thus, U-Net outperformed the other models in tobacco seedling segmentation. Although the U-Net model had lower Precision than Linknet and lower Recall than U-Net++, it achieved the best scores for the mIoU (90.70%), PA (98.61%), and Kappa Coefficient (0.9442) metrics; thus, the U-Net model was best suited to various scenarios of tobacco seedling segmentation. The segmentation performance of PSPNet was much lower than that of the other seven models, with mIoU, PA, Precision, Recall, and Kappa Coefficient values of 80.49%, 97.22%, 94.41%, 87.14%, and 0.8308, respectively. Figure 6 presents the comparison results of the performance of the eight different DL models in segmenting tobacco plants in the hilly dam area.

#### 3.1.4. Model Segmentation Efficiency

Table 7 summarizes the training time of the eight deep learning models: U-Net, U-Net++, Linknet, PSPNet, MAnet, FPN, PAN, and DeepLabV3+. MAnet took the longest time to train at 171.54 min, with all the other parameters being equal. Meanwhile, the FPN model had the shortest training time of 86.1 min, indicating that FPN is the easiest and computationally cheapest model to train and is the best model for tobacco seedling segmentation in terms of speed. Although the training time of PAN was not the shortest, it was second only to FPN and shorter than that of the U-Net, U-Net++, Linknet, PSPNet, Manet, and DeepLabV3+ models. The U-Net and U-Net++ models took a relatively long training time of 122.75 min and 100.27 min, respectively; thus, they were more time-consuming and failed to outperform the other models.

### 3.2. Visualization and Assessment of the Tobacco Seedling Segmentation Results

#### 3.2.1. Visual Evaluation of the HSTD Training Model Segmentation Results

This study explored the problems and advantages of U-Net, U-Net++, Linknet, PSPNet, MAnet, FPN, PAN, and DeepLabV3+ through the visual evaluation of the segmentation results of each model and obtained an optimal model for tobacco seedling segmentation in the hilly areas of the Karst mountains. Figure 7 shows the visualization results of the eight deep learning models, including the partial segmentation results of eight different scenarios, denoted by the numbers I–VIII.

As shown in Figure 7, the overall performance of the eight deep learning models in segmenting tobacco seedlings was better, with good segmentation results for flat–fragmented plots, plots with good–poor tobacco seedling growth, and plots with more–less weeds. However, there were still a high number of tobacco plant seedlings that were omitted in the splitting process or incorrect splitting of other objects into tobacco plants, as well as a high number of incomplete splits of whole tobacco plants: (1) Missed segmentation: this problem affected tobacco plants of a small size more; the PSPNet model especially showed greater missed segmentation (Figure 7I,VI). In addition, tobacco plant missed segmentation was more severe with ridge shading, with the most affected models being U-Net, U-Net++, PSPNet, and Deeplabv3+, while Linknet and FPN were affected to a lesser extent (Figure 7IV,VIII). Furthermore, there was more missed segmentation of tobacco plants when the plots were fragmented, with the greatest impact on U-Net++ and PSPNet (Figure 7V); (2) Mis-segmentation situation: the MAnet, FPN, PAN, and Deeplabv3+ models showed more mis-segmented results due to the influence of weed color (Figure 7I,VII,VIII); (3) Incomplete segmentation of tobacco plants: because of the plant size, background brightness, and plot fragmentation, tobacco plants could be incompletely segmented, which affected the segmentation accuracy of the models (Figure 7II,III,V). By analyzing the segmentation results of the eight models, it was found that the models with the worst segmentation accuracy, PSPNet and PAN, had problems with tobacco plant segmentation in all eight scenarios; when the seeds are large, they should be segmented as single tobacco seedlings, but these two models segmented multiple plants as one plant, which directly affected the accurate counting of tobacco plants (Figure 7I,II,IV,VII,VIII).

Overall, the segmentation performances of U-Net and U-Net++ were the best, while those of PSPNet and PAN was poor. Meanwhile, weeds were the main influencing factor of model mis-segmentation, and the small sizes of the tobacco plants and shading were the main influencing factors of model missed segmentation.

#### 3.2.2. Visualization and Evaluation of the DASTD Training Model Segmentation Results

Figure 8 illustrates the segmentation results of U-Net, U-Net++, Linknet, PSPNet, MAnet, FPN, PAN, and DeepLabV3+ for tobacco seedlings in the dam area. Compared with the segmentation of the tobacco seedlings in the hilly area (Figure 7), the planting environment in the dam area is simpler, with flat plots, fewer weeds in the tobacco fields, and regular tobacco planting; therefore, all eight semantic segmentation models showed better overall segmentation results for the dam area. Visually, U-Net outperformed the other models in distinguishing between tobacco and non-tobacco seedlings without excessive confusion and misclassification; all the regions were accurately segmented, except for some missed segmentation (Figure 8IV) due to tree shading and some mis-segmentation (Figure 8IV,VI) due to the influence of trees and weeds. The other models showed less accurate segmentation results compared with U-Net: (1) For the case of missed segmentation, the models were affected by the shade of trees and the small size of tobacco plants, with the main influencing factor being the shade of trees (Figure 8IV,VI,VIII); the most affected models were the PSPNet and PAN models, where none of the tobacco plants that were shaded by trees were segmented out (Figure 8IV); (2) The case of mis-segmentation was influenced by the presence of a small number of weeds and trees in the tobacco field, which the models mis-segmented as tobacco plants (Figure 8II–VIII).

In conclusion, based on the above observations, U-Net was better able to segment the tobacco seedlings from the whole image, especially in avoiding the misclassification of trees and weeds. In contrast, PSPNet segmented more weeds into tobacco plants in the tobacco field, resulting in the worst segmentation performance. Meanwhile, weeds posed greatest influence on the segmentation of tobacco plants in the dam area.

#### 3.2.3. Visualization and Evaluation of the H-DASTD Training Model Segmentation Results

The accuracy of U-Net, U-Net++, Linknet, PSPNet, MAnet, FPN, PAN, and DeepLabV3+ trained on the H-DASTD in segmenting tobacco seedlings in the hilly and dam areas was analyzed. Figure 9 shows a visualization map of the segmentation results in the dam (Figure 9I–IV) and hilly areas (Figure 9V–VIII) to more clearly observe the performance of each model.

Comparing the segmentation results for the dam and hilly areas, it was found that although the various parameter models were the same, the prediction results for the dam and hilly areas showed a big difference: (1) The division of tobacco plants in the dam area was complete, and there were few cases of misdivision (Figure 9I–IV), while in the hilly area, the tobacco plant segmentation was incomplete, there were more fine weed patches, and the performances of PSPNet and PAN were the best (Figure 9VI,VII). At the same time, due to the better growth of some tobacco plants in the hilly area, the gap between plants was small, and there was incomplete segmentation of multiple plants into a single plant, which was especially obvious in the segmentation results of the PSPNet and PAN models (Figure 9VII,VIII); (2) In the case of plants of the same length and size (Figure 9II,V), the results of splitting were clear at a glance and wrongly split tobacco plants could be quickly observed due to the flatness of the plots in the dam area, whereas in the hilly area, the plots were broken up, there was a large number of weeds, and, therefore, the results of splitting were observed to be haphazard.

A comparison of the eight models revealed that U-Net was the model with a good cutting performance, achieving good segmentation results in both the dam and hilly areas, with the complete segmentation of tobacco plants and a small number of mis-segmentations (Figure 9IV,VIII). Compared with U-Net, the performance of U-Net++ was relatively poor with more wrong and omitted segments, but it still achieved better results than the other models, with the complete extraction of tobacco plants (Figure 9V,IV,VIII). Among the other models, FPN, PSPNet, and PAN showed the poorest segmentation performance, especially in the complex hilly area, where a large number of missed and incomplete segmentations of tobacco plants were observed.

## 4. Discussion

### 4.1. Analysis of Errors and Omissions of Deep Learning Models in Dam and Hilly Areas

This study analyzed the results of U-Net, U-Net++, Linknet, PSPNet, MAnet, FPN, PAN, and DeepLabV3+ models in segmenting tobacco plants in the dam and hilly areas (Figure 10).

(1) The analysis of the missed segmentation of tobacco seedlings revealed (Figure 10a) that the number of tobacco seedlings that were missed in the dam area was much less than the number of tobacco seedlings that were missed in the hilly area. Among the eight semantic segmentation models, the one with the smallest number of missed segmentations in the dam area was the U-Net++ model, which missed only 135 plants, and the one with the highest number of missed segments was the PSPNet model, with 878 plants. However, when comparing the results of tobacco plant seedling missed segmentation in the hilly area, PSPNet also achieved better results in the dam area; the model with the lowest missed segmentation rate in the hilly area was the FPN model, with 2813 plants being missed, while the missed segmentation of PSPNet was the highest, with 12,344 plants being missed, showing a difference of 9531 plants; thus, the difference in the performances of FPN and PSPNet in the hilly area was extremely large, indicating that the PSPNet model is greatly affected by complex terrain, broken plots, and the crop-planting environment, and it is not applicable for tobacco plant seedling segmentation in complex hilly areas. The models with the next worse performance were PAN and DeepLabv3+, with 5974 and 4989 plants being missed in the hilly area and 567 and 645 plants being missed in the dam area, respectively. However, the segmentation performances of PAN and DeepLabv3+ were still better than that of PSPNet, while a better segmentation performance was achieved by U-Net, U-Net++, Linknet, MAnet, and FPN, with 2867, 2986, 2933, 3164, and 2813 plants missed in the hilly area and 149, 135, 134, 147, and 194 plants missed in the dam area, respectively, implying that U-Net, U-Net++, Linknet, MAnet, and FPN are more applicable to the segmentation of crop seedlings in dam areas.

(2) The analysis of misclassified tobacco seedlings revealed (Figure 10b) that the mis-segmentation of tobacco seedlings in the hilly area was much higher than in the dam area. PSPNet showed the smallest difference of 341 plants, while PAN showed the largest difference of 6568 plants. Comparing the eight models, the U-Net++ segmentation performance was the best in the dam area, with only 387 mis-segmentations, followed by U-Net (526), MAnet (637), and then DeepLabv3+ (930), whereas Linknet (1179), FPN (1108), and PSPNet (1321) had a comparable number of mis-segmentations, suggesting that they segmented equally well in the dam area, while PAN had the highest number of 1321 mis-segmentations in the dam area. When analyzing the mis-segmentation problem in the hilly area, the model with the lowest mis-segmentation rate was PSPNet, with only 1662 plants being mis-segmented; the highest mis-segmentation rate was observed for PAN, with 9871 plants being mis-segmented. As shown in the figure, the model segmentation performance is in the order of PSPNet (1662 plants) > Linknet (1767 plants) > U-Net (3024 plants) > U-Net++ (3882 plants) > MAnet (4626 plants) > FPN (5617 plants) > DeepLabV3+ (5644 plants) > PAN (9871 plants). Among the eight segmentation models, Linknet and PSPNet had a comparable number of wrong segmentations in the dam and hilly areas, indicating that topography and parcel fragmentation do not have much influence on them; the other models are more influenced by factors such as topography and parcel fragmentation.

In summary, for tobacco plant seedling segmentation, topography and land fragmentation have a large impact on model segmentation, with missed segmentation of tobacco plant seedlings in dams being far lower than in hilly areas. The U-Net model has the lowest missed segmentation rate in the dam and hilly areas, while PSPNet has the highest missed segmentation rate. In the dam area, U-Net++ mis-segmentation is the lowest and PAN mis-segmentation is the highest, while, in the hilly area, PSPNet mis-segmentation is the lowest and PAN mis-segmentation is the highest.

### 4.2. Analysis of Factors Affecting Model Misclassification in the Dam Area

In the Karst region, the dam area is an area with a relatively flat topography with contiguous plots. The performance of different models in segmenting tobacco seedlings in the dam area was analyzed in detail to identify the influencing factors of model mis-segmentation.

Figure 11 illustrates the influencing factors of missed segmentation and their percentage for different models in the dam area. As shown in Figure 11a, in the dam area, PSPNet was the model with the highest missed segmentation rate of 30.82%, followed by PAN and DeepLabv3+ with rates of 19.90% and 22.64%, respectively, whereas U-Net, U-Net++, Linknet, MAnet, and FPN showed lower missed segmentation rates in the dam area. As shown in Figure 11b, in the dam area, the two factors affecting model missed segmentation were small tobacco plants (STPs) and tree cover (TC), with STPs being the main influencing factor. There were some differences in the degree of influence of the factors on the different models, with STPs having the greatest influence on DeepLabv3+, with a missed segmentation rate of 88.4%, and the least influence on Linknet, with a missed segmentation rate of 78.2%.

Figure 12 illustrates the influencing factors of mis-segmentation and their proportions for the different models in the dam area. As shown in Figure 12a, in the dam area, PAN was the model with the highest mis-segmentation rate among the eight models, accounting for 35.17% of all mis-segmentations, while the mis-segmentation rates of PSPNet, Linknet, FPN, and DeepLabv3+ were relatively lower, accounting for 14.07%, 12.55%, 11.8%, and 9.90%, respectively. MAnet and U-Net mis-segmentation accounted for 6.78% and 5.6% of all mis-segmentations, while U-Net++ had the lowest mis-segmentation rate at 4.12%. As shown in Figure 12b, there were five types of objects affecting model mis-segmentation in the dam area, namely, weeds, corn plants (CPs), trees, chili pepper plants (CPPs), and others. The most mis-segmented object was weeds, with Linknet showing the highest mis-segmentation rate of 80.66%, and the lowest rate was shown by PSPNet at 57.31%. U-Net, U-Net++, MAnet, Linknet, and PFN mis-segmented more trees, and PSPNet, PAN, and DeepLabv3+ mis-segmented more CPs; meanwhile, CPPs and other objects were rarely mis-segmented.

In summary, the main factor affecting model missed segmentation in the dam area was STP, while the main factor affecting model mis-segmentation was weeds.

### 4.3. Factors Affecting Model Misclassification in Hilly Areas

In the Karst region, the hilly area is an area with high topographic relief and fragmented plots. This study analyzed the performance of different models in segmenting tobacco seedlings in the hilly area in detail.

Figure 13 illustrates the missed segmentation of different models for tobacco seedlings in the hilly area. As can be seen in Figure 13a, PSPNet was the model with the highest missed segmentation rate among the eight models, accounting for 32.42%, followed by PAN and DeepLabv3+, with a missed segmentation rate of 15.69% and 13.10%, respectively. The U-Net, U-Net++, Linknet, Manet, and FPN mis-segmentation rates were comparable, being in the range of 7.39% to 8.31%. As shown in Figure 13b, the factors affecting missed segmentation in the hilly area were shadows, numerous weeds (NW), STP, and image blur (IB). Among them, the most influential factor was STP, with the eight models showing a missed segmentation rate of STP between 54.41% and 74.74%, followed by NW with a rate between 13.93% and 27.55%. The mis-segmentation rate of shadows was between 5.28% and 17.11%, and the least influential factor was IB, with a rate between 1% and 2.3%. In summary, the main influencing factor of model mis-segmentation was STP.

Figure 14 illustrates the mis-segmentation of different models in the hilly area. As can be seen in Figure 14a, PAN had the highest mis-segmentation rate in the hilly area, accounting for 27.35% of all mis-segmentations; PSPNet showed the lowest mis-segmentation rate, accounting for 4.6% of all mis-segmentations. As shown in Figure 14b, the factors affecting model mis-segmentation in the hilly area were weeds, CP, rocks, and others; among them, weeds were the main factor affecting model mis-segmentation, with the largest influence on Linknet, accounting for 91.62% of all mis-segmentations, and the lowest influence on PAN, accounting for 68.32% of all mis-segmentations. CP is the next most influential factor, with the greatest impact on PAN, accounting for 23.96% of all mis-segmentations, and the lowest influence on Deeplabv3+, accounting for 4.22% of all mis-segmentations. Rocks had the greatest influence on U-Net, accounting for 10.52% of all mis-segmentations, and the lowest influence on PSPNet, accounting for 2.83%.

In summary, the main factors affecting model mis-segmentation in the hilly area were STP, shadows, and NW; meanwhile, the main factor affecting model mis-segmentation was weeds.

### 4.4. Differences from Existing Research

The results of this study indicate that semantic segmentation models can effectively segment tobacco seedlings when trained on low-altitude drone images. Considering the challenging conditions created by the topography and landforms of Karst mountainous areas (e.g., dam areas and hilly areas), this study found that semantic segmentation models performed better in segmenting tobacco seedlings in the dam area than in the hilly area. It is difficult to consistently segment tobacco seedlings in the hilly area because of the complex plots, significant differences in the growth of tobacco seedlings, and the diversity of tobacco seedling plants in the labeled images. Compared with tobacco seedlings in the dam area, the tobacco contours are less distinct and there are more weeds. Although including the annotation of a large number of tobacco seedling samples when training models can help to increase sample diversity and improve segmentation accuracy, terrain fragmentation in hilly areas still has an impact on the model segmentation of tobacco seedlings.

Although there are few studies that show direct similarities with this study, we comment on other works in the relevant literature and how they compare with our study. Liang Huang et al. [8] used DeeplabV3+, PSPNet, SegNet, and U-Net to segment tobacco planting areas in high-altitude mountainous areas, and obtained mIoU values of 0.9436, 0.9118, 0.9392, and 0.9473, respectively, verifying the feasibility of using semantic segmentation models to extract tobacco planting areas from drone remote-sensing images. Xiaodong Bai et al. [42] designed an RPNet model to count rice plants after the tillering stage and compared this model with MCNN, CSRNet, SANet, TasselNetV2, and FIDTM. The MAE, RMSE, rMAE, and rRMSE values increased by 8.3%, 11.2%, 1.2%, and 1.6%, respectively. Therefore, RPNet can be used to accurately count rice plants in paddy fields, replacing traditional manual counting methods. Jie Li et al. [43] designed RapeNet and RapeNet+ models for the automatic counting of rapeseed flower clusters. The experimental results showed that the RapeNet series outperformed other state-of-the-art counting methods, providing important technical support for field rapeseed flower cluster counting statistics. In the existing research, Liang Huan et al.’s method is the most similar to ours, and although their method segments different objects, it uses the same model for segmentation and achieves better segmentation accuracy.

### 4.5. Limitations of the Study

This study used semantic segmentation models to segment tobacco seedlings in dams and hilly areas. Although good results were achieved, there are still limitations, mainly in the following aspects:

(1) The terrain in Karst mountainous areas is fragmented, and the composition of near-ground objects is complex, with staggered power lines, signal base stations, and other objects. It is difficult for drones to collect high-resolution images by flying at ultra-low altitudes close to the ground. Due to interference factors such as meteorology, topography, and signal base stations that affect the safe flight of drones, the spatial resolution of images may not be able to meet the precise segmentation of tobacco seedlings. For practical applications, factors such as the spatial resolution of data, the planting structure of the research area, and the fragmentation of farmland should be considered to determine the optimal monitoring time and required image spatial resolution, as well as to develop the corresponding segmentation method with the lowest cost to meet the accuracy requirements for segmenting tobacco seedlings.

(2) The results of this study are based only on drone images from two areas: a dam area and a hilly area. Although this study selected tobacco seedling planting sites based on the topography and landforms of Karst mountainous areas, it cannot guarantee that these sites fully represent other tobacco planting areas in Karst mountainous areas. Similarly, obtaining high-resolution drone images is difficult, and a large number of tobacco seedling plant samples cannot be obtained. Although various processing methods were applied to the samples to increase their diversity, they cannot represent all types of tobacco cultivation in Karst mountainous areas, which affects model training and reduces the performance of model segmentation.

## 5. Conclusions

Considering the challenging conditions (e.g., intercropping, fragmented plots, weeds, trees, shadows, and soil canopies) posed by the topography of the Karst mountainous region and the environment in which crops are grown, this study chose two areas—namely, a dam area and a hilly area—for the precision segmentation of tobacco seedlings. It was found that the semantic segmentation models (U-Net, U-Net++, Linknet, PSPNet, MAnet, FPN, PAN, and DeepLabV3+) trained on low-altitude UAV RGB imagery could efficiently segment tobacco seedlings grown in the complex environments of Karst mountainous areas (dams and hilly areas).

(1) Topography and geomorphology have a greater impact on the model segmentation of tobacco seedlings, and the accuracy of the model segmentation results for tobacco seedlings in the dam area is much better than the accuracy of the segmentation results in the hilly area. U-Net, U-Net++, Linknet, PSPNet, MAnet, FPN, PAN, and DeepLabV3+ show excellent results in segmenting tobacco seedlings in the dam area, with mean mIoU, PA, Precision, Recall, and Kappa Coefficient values reaching 87%, 97%, 91%, 85%, and 0.81, respectively. In the hilly area, U-Net, U-Net++, Linknet, PSPNet, MAnet, FPN, PAN, and DeepLabV3+ segmentation are relatively poor, with the mean mIoU, PA, Precision, Recall, and Kappa Coefficient values for these models being 81.04%, 97.97%, 72.64%, 73.47%, and 0.73, respectively.

(2) The tobacco plant seedling segmentation performance of different models varied widely. Overall, in the hilly area, the segmentation performances of U-Net, U-Net++, Linknet, and PSPNet were better than those of MAnet, FPN, PAN, and DeepLabV3+, with higher mIoU, PA, Precision, Recall, and Kappa Coefficient values of 83.35%, 98.38%, 77.26%, 79.26%, and 0.7799 for U-net and 84.17%, 98.56%, 75.84%, 80.13%, and 0.7766 for U-Net++, respectively. In the dam area, the overall model segmentation accuracy was superior, with higher values associated with better model performance, according to the mIoU, with U-Net (93.83%) > U-Net++ (92.90%) > Linknet (92.78%) > MAnet (92.74%) > FPN (92.57%) > DeepLabV3+ (92.22%) > PAN (90.02%) > PSPNet (87.78%), where U-Net had the best overall performance and PSPNet had the worst performance. Therefore, U-Net is well-suited for crop segmentation in dams with simple cropping environments and can provide effective modeling applications for various crop segmentation tasks.

(3) Through an analysis of the performance of U-Net, U-Net++, Linknet, PSPNet, MAnet, FPN, PAN, and DeepLabV3+ in tobacco segmentation in the dam and hilly areas, it was found that factors such as topography and land fragmentation had significant impacts on model errors and omissions during segmentation. U-Net, U-Net++, Linknet, MAnet, and FPN showed a better segmentation performance, with missed segmentation of 2867, 2986, 2933, 3164, and 2813 plants in the hilly area and 149, 135, 134, 147, and 194 plants in the dam area, respectively. This means that U-Net, U-Net++, Linknet, MAnet, and FPN are more suitable for the segmentation of crop seedlings in dam areas. The misclassification of tobacco seedlings in the hilly area was much higher than in the dam area, with PSPNet showing the smallest difference at 341 plants, while PAN showed the largest difference at 6568 plants. Among the eight segmentation models, Linknet and PSPNet had similar numbers of misclassifications in the dam and hilly areas, and the terrain, landforms, and fragmentation of the land parcels had little impact on them. U-Net, U-Net++, MAnet, FPN, PAN, and DeepLabV3+ were greatly affected by factors such as terrain, landforms, and land fragmentation. Therefore, for Karst mountainous areas with diverse landforms, accurate crop segmentation was mainly influenced by terrain and landforms, the effects of which cannot be generalized and require zoning research.

(4) Among the different models, the U-Net model had the lowest mis-segmentation rate in both the dam and hilly areas, while PSPNet had the highest mis-segmentation rate. In the dam area, U-Net++ had the lowest mis-segmentation rate and PAN had the highest mis-segmentation rate while, in the hilly area, PSPNet had the lowest mis-segmentation rate and PAN had the highest mis-segmentation rate. In addition, the factors affecting model segmentation differed between the dam area and the hilly area; the factors affecting model missed segmentation in the dam area were few tobacco plants and trees, with STP being the main influencing factor. The factors affecting model mis-segmentation included weeds, CP, trees, and CPP, among others, with weeds, CP, and trees being the main influencing factors. In the hilly area, shadows, NW, STP, and IB were the factors affecting model missed segmentation, with STP, shadows, and NW being the main factors; weeds, CP, rocks, and others were the factors affecting model mis-segmentation, with weeds being the main influencing factor.

## Figures and Tables

**Figure 1 plants-13-03186-f001:**
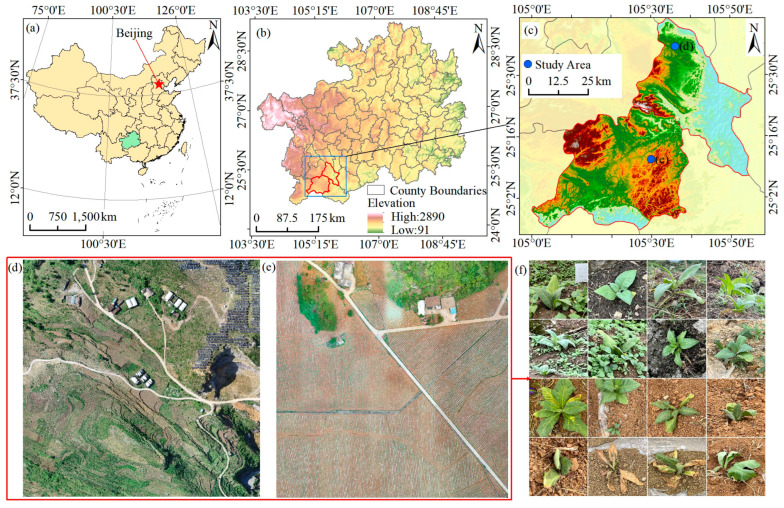
Study area map. (**a**) China; (**b**) Guizhou province; (**c**) Zhenfeng and Anlong Counties; (**d**) UAV imagery data of the study area in the Zhenfeng and (**e**) Anlong Counties; (**f**) planting environments, growth, management of pests and diseases, and replanting of tobacco plants in the two study areas.

**Figure 2 plants-13-03186-f002:**
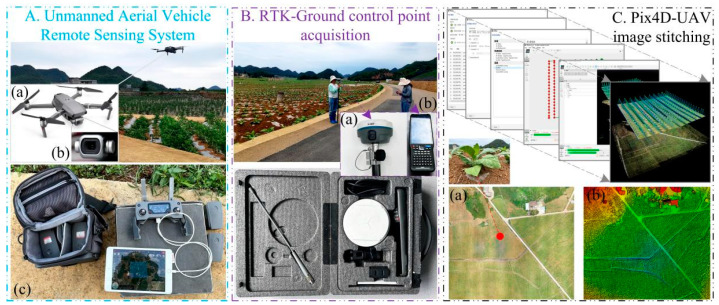
Acquisition and stitching of drone data. (**A**) UAS: (**a**) DJI Mavic 2 Pro drone equipped with (**b**) Hasselblad camera and (**c**) drone batteries, handle, tablet, etc.; (**B**) RTK ground control point data acquisition: (**a**) satellite signal receiver and (**b**) smart handbook (interactive interface); (**C**) Drone image stitching software—Pix4D: (**a**) Digital Orthophoto Map (DOM) and (**b**) Digital Surface Model (DSM).

**Figure 3 plants-13-03186-f003:**
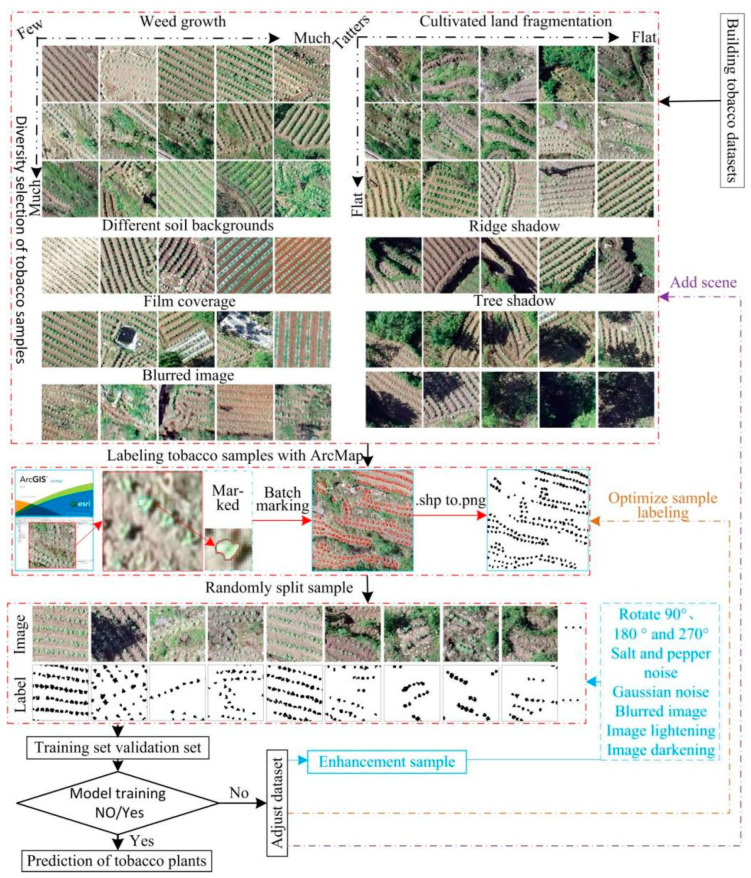
Data set construction process.

**Figure 4 plants-13-03186-f004:**
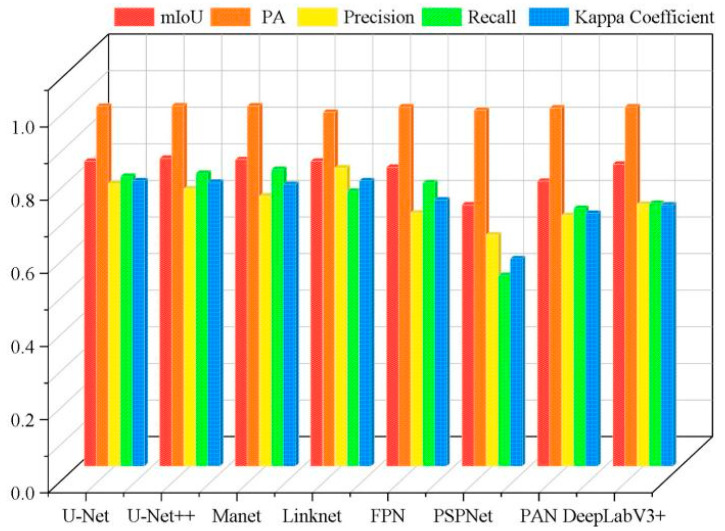
Histogram of the accuracy of the different models in the hilly area.

**Figure 5 plants-13-03186-f005:**
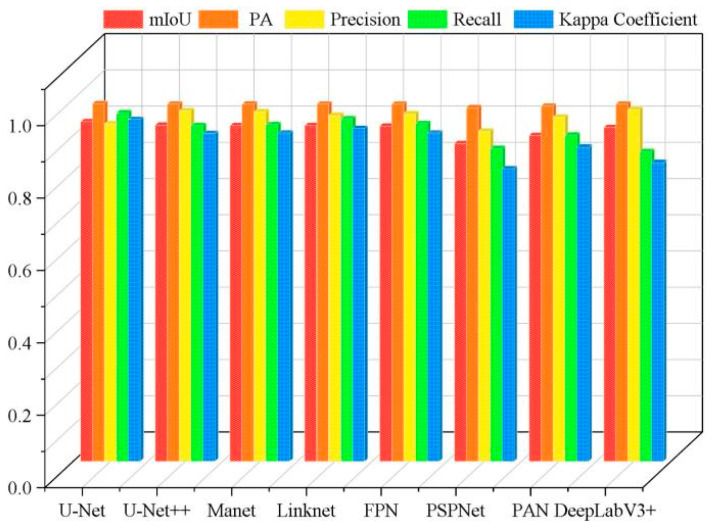
Histogram of the accuracy of the different models in the dam area.

**Figure 6 plants-13-03186-f006:**
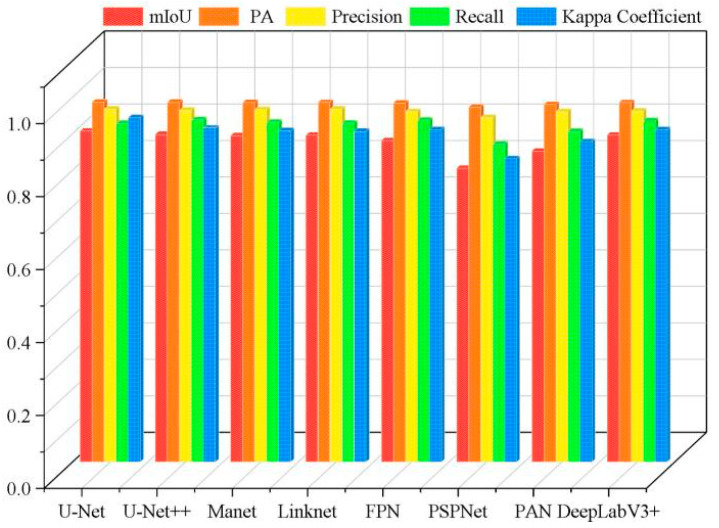
Histogram of the accuracy of the different models in the hilly dam area.

**Figure 7 plants-13-03186-f007:**
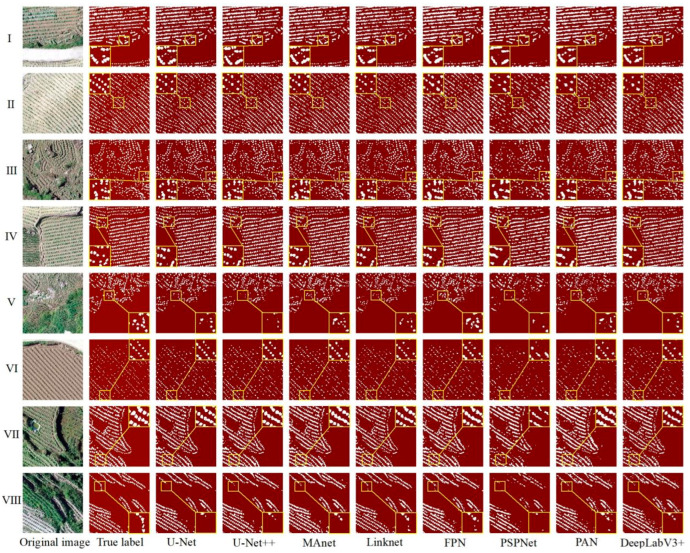
Visualization of the segmentation results of the eight deep learning models trained on the HSTD. I—tobacco plants of different sizes and the plots are flat; II—small tobacco plants and the plots are flat; III—small tobacco plants and the plots are more fragmented; IV—large tobacco plants and the plots are flat; V—large tobacco plants and the plots are fragmented; VI—small tobacco plants and the plots are flat; VII—large tobacco plants, the plots are flat with more weeds in the background, and the ridges of the soil are wide, high, and shaded; VIII—large tobacco plants, the plots are broken, and the ridges are wide, high, and shaded.

**Figure 8 plants-13-03186-f008:**
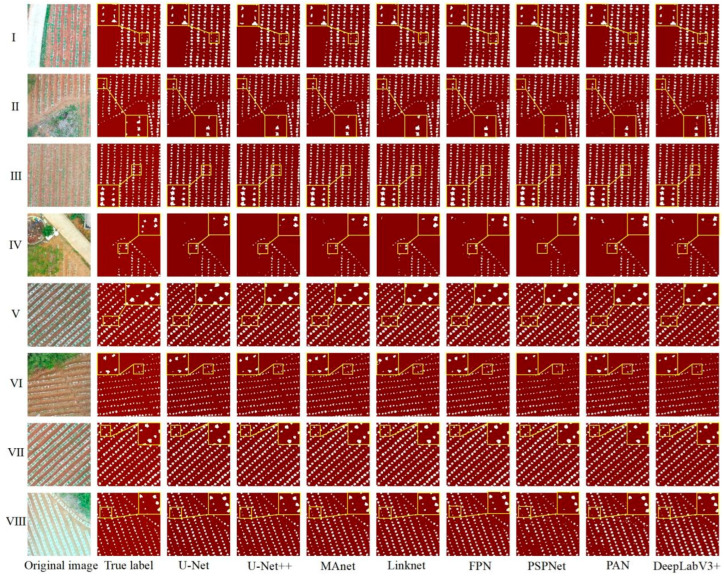
Visualization of the segmentation results of the eight deep learning models trained on the DASTD. I—presence of road, lots of weeds, and large tobacco plants; II—presence of trees, small amount of weeds, and small tobacco plants; III—small tobacco plants; IV—presence of trees, road, and other debris; V—large tobacco plants; VI—presence of trees and small tobacco plants; VII—large tobacco plants; VIII—presence of large trees, large tobacco plants, and brighter light in the region.

**Figure 9 plants-13-03186-f009:**
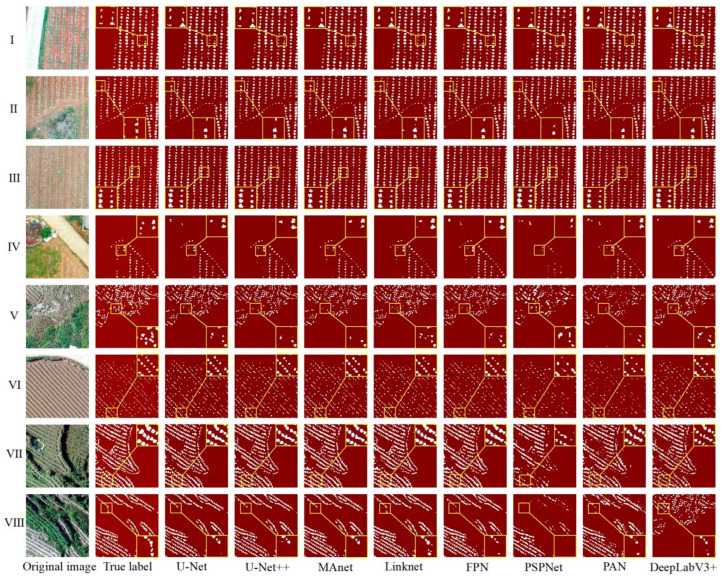
Visualization of the segmentation results of the eight deep learning models trained on the H-DASTD. I—presence of road, lots of weeds, and large tobacco plants; II—presence of trees, small amount of weeds, and small tobacco plants; III—small tobacco plants; IV—presence of trees, road, and other debris; V—large tobacco plants and fragmented plot; VI—small tobacco plants and flat plot; VII—large tobacco plants, flatter plot, more weeds in the background, and soil ridge is wide and high with shading; VIII—large tobacco plants, fragmented plots, and wide and high ridges with shading.

**Figure 10 plants-13-03186-f010:**
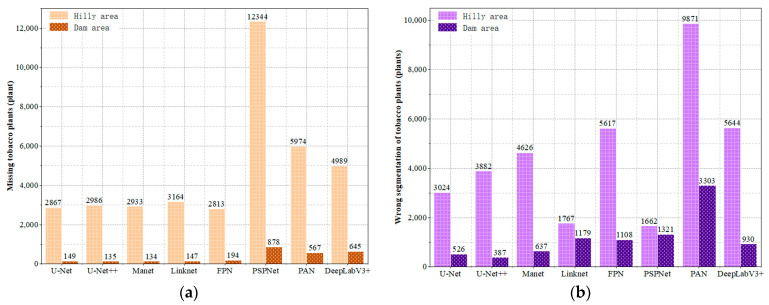
Histograms of mis-segmented tobacco plants in dam and hilly areas: (**a**) histogram of mis-segmented tobacco plants; (**b**) histogram of mis-segmented tobacco plants.

**Figure 11 plants-13-03186-f011:**
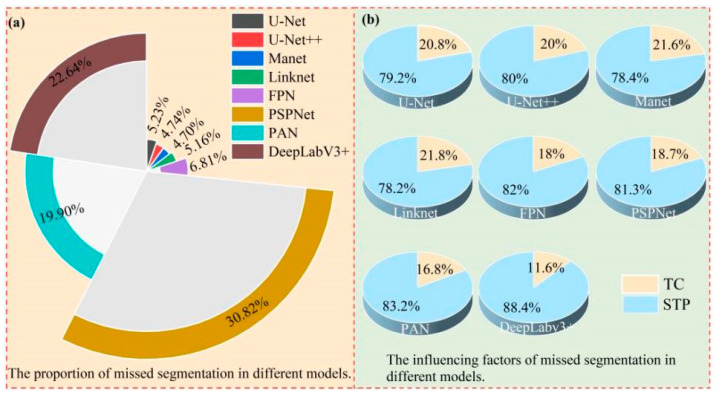
Map of factors affecting missed segmentation for different models in the dam area.

**Figure 12 plants-13-03186-f012:**
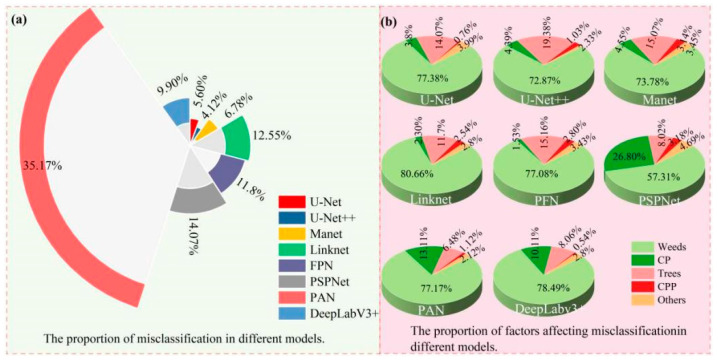
Map of factors influencing the mis-segmentation of different models in the dam area.

**Figure 13 plants-13-03186-f013:**
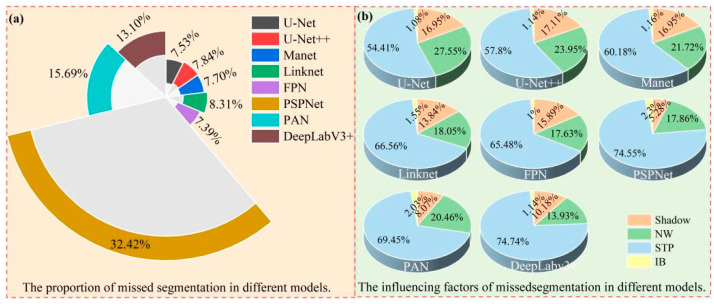
Map of factors affecting the mis-segmentation rates of different models in the hilly area.

**Figure 14 plants-13-03186-f014:**
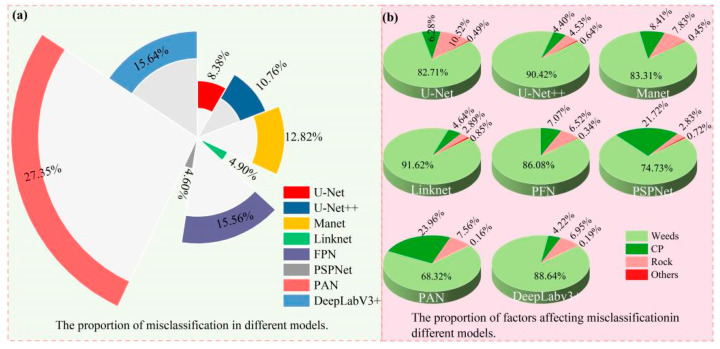
Map of factors influencing the mis-segmentation of different models in the hilly area.

**Table 1 plants-13-03186-t001:** Detailed descriptions of HSTD, DASTD, and H-DASTD.

Datasets	Tobacco Seedling Growing Environment	Number of Samples of Tobacco Seedling Plants	Partial Sample Display (Image)
HSTD	The planting land has a Karst hilly landscape, with large surface undulation, broken cultivated land, diverse soils, high and wide soil cans, uncovered membranes, many weeds in the background of tobacco planting, and different growth rates of tobacco; in addition to tobacco planting, a large amount of maize and a small amount of rice and vegetables are also planted.	Manually labeled samples of 24,558 plants: the labeled tobacco seedling images were binarized and segmented into 224 pixel × 224 pixel-sized slices, and geometric and color transformations were performed on the samples to obtain rich and diverse tobacco samples, totaling 1800 samples.	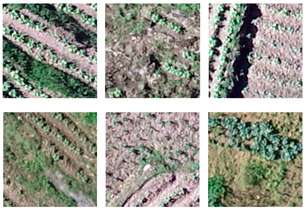
DASTD	The planting site is in a dam area, with regular tobacco planting, flat cultivated fields, no soil banks, monoculture soil, mulch cover, and varying tobacco growth; the major planting crop is tobacco, with a small amount of maize and chili peppers also planted.	Manually labeled samples of 32,261 plants: the labeled tobacco seedling images were binarized and segmented into 224 pixel × 224 pixel-sized slices, and geometric and color transformations were performed on the samples to obtain rich and diverse tobacco samples, totaling 1900 samples.	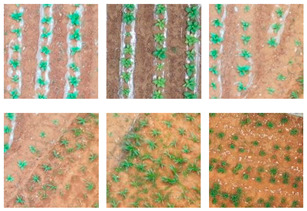
H-DASTD includes all tobacco samples from both the HSTD and DASTD, with a total of 3700 samples.			

**Table 2 plants-13-03186-t002:** Detailed overview of the eight deep learning models.

Model	Modeling Framework	Presentation and Application of the Model
U-Net	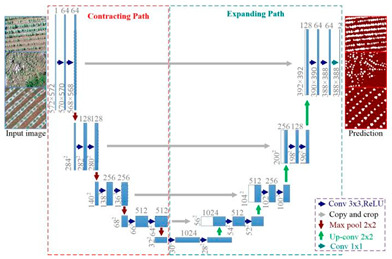	U-Net was proposed by Olaf Ronneberger et al. [34] to solve the problem of biomedical image segmentation. As the model is used for small sample segmentation with a good effect, it has been widely used in various fields, especially since the field of remote sensing has greatly developed.
U-Net++	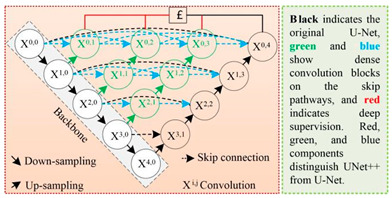	U-Net++ was proposed by Zongwei Zhou et al. [35] to solve the accuracy problem of medical image segmentation, aiming to reduce the semantic gap between the feature maps of the encoder and decoder subnetworks. It was later widely used in the image segmentation of objects such as roads, water bodies, crops, and buildings.
Linknet		Linknet is a framework proposed by Abhishek Chaurasi et al. [36] to address the efficiency of pixel-by-pixel semantic segmentation of visual scenes; the architecture allows learning without significantly increasing the number of parameters, obtaining a high segmentation accuracy without affecting the processing time, and can be used for different image resolution segmentation applications, including roads, rivers, crops, buildings, and fruits.
PSPNet	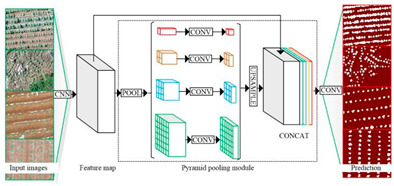	PSPNet was proposed by Hengshuang Zhao et al. [37] to solve the problem of scene analysis. PSPNet corrects the problems of the FCN network on scene analysis datasets such as mismatching, the confusion of segmentation results, and the loss of small targets, which enhances the accurate segmentation of objects of similar color and shape in a scene analysis. It is now applied to objects such as arable land, water bodies, forests, animals, crops, fruits, roads, and buildings.
MAnet	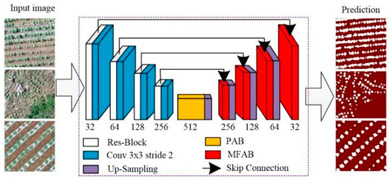	MAnet is a model proposed by Tongle Fan et al. [38] as a multi-scale attention network for liver and tumor segmentation to solve the problems that null convolution and pooling operations cannot exploit, the spatial and channel relationships between pixels in the global view, and pooling operations can easily lose the details from the feature map information. It has later been used for face recognition, expression recognition, mineral segmentation, and so on.
FPN	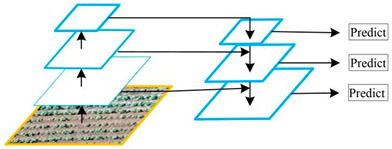	FPN is a model proposed by TAlexander Kirillov et al. [39] to deal with the deficiencies of the multiscale variability problem, reduce the computational time, and improve the model performance.
PAN	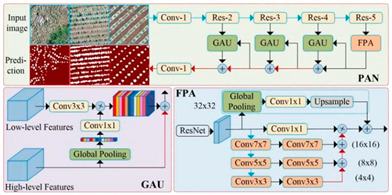	PAN was proposed by Hanchao Li et al. [40] to solve the problems of the difficulty in classifying small categories and time-consuming model training on multiple scales. This model combines an attention mechanism and spatial pyramid to extract precise and dense features for pixel labeling, which can effectively increase the sensory field and classify small objects.
DeepLabV3+	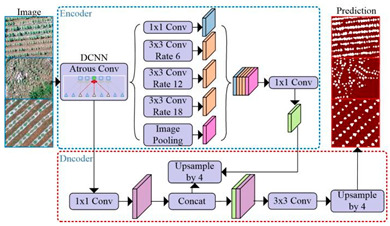	DeepLabV3+ is a model proposed by Liang Chieh Chen et al. [41] to solve the problem that the multiple downsampling of DCNN will cause the resolution of feature maps to become smaller, resulting in a lower prediction accuracy, the loss of boundary information, and excessive time consumption. It is now widely used in the segmentation of crops, forests, land cover information, roads, buildings, water bodies, tea gardens, weeds, and so on.

**Table 3 plants-13-03186-t003:** Details on the hardware and software environments.

Basic Configuration Project	Basic Configuration Content	Software Configuration Project	Software Version
System	Windows 10 professional edition	GPU-Drive	560.94
CPU	13th Gen Intel(R) Core(TM) i7-13700KF 3.40 GHz	CUDA	12.6
RAM	32 GB	Python	3.7
Hard disk	1940 GB	Tensorflow	2.0.0
Graphics card	NVIDIA GeForce RTX4080	Torch	1.7.1

**Table 4 plants-13-03186-t004:** Accuracy of the segmentation results of the different DL models in the hilly area.

Model	mIoU/%	PA/%	Precision/%	Recall/%	Kappa Coefficient
U-Net	83.35	98.38	77.26	79.26	0.7799
U-Net++	84.17	98.56	75.84	80.13	0.7766
MAnet	83.77	98.47	73.85	81.14	0.7704
Linknet	83.40	96.74	81.55	75.21	0.7800
FPN	81.68	98.21	69.20	77.52	0.7278
PSPNet	71.46	97.27	63.23	52.16	0.5670
PAN	77.90	97.90	68.53	70.49	0.6913
DeepLabV3+	82.62	98.29	71.63	71.88	0.7142

**Table 5 plants-13-03186-t005:** Accuracy of the segmentation results of the different DL models in the dam area.

Model	mIoU/%	PA/%	Precision/%	Recall/%	Kappa Coefficient
U-Net	93.83	98.83	93.27	96.24	0.9440
U-Net++	92.90	98.70	96.83	92.78	0.9055
MAnet	92.74	98.66	96.56	93.00	0.9066
Linknet	92.78	98.70	95.57	94.63	0.9194
FPN	92.57	98.64	95.99	93.25	0.9066
PSPNet	87.78	97.66	91.18	86.44	0.8080
PAN	90.02	98.16	95.07	90.14	0.8677
DeepLabV3+	92.22	98.65	97.21	85.58	0.8251

**Table 6 plants-13-03186-t006:** Accuracy of the segmentation results of the different DL models in the hilly dam area.

Model	mIoU/%	PA/%	Precision/%	Recall/%	Kappa Coefficient
U-Net	90.70	98.61	96.74	92.82	0.9442
U-Net++	89.80	98.60	96.44	93.82	0.9150
MAnet	89.46	98.53	96.59	93.15	0.9084
Linknet	89.58	98.57	96.75	92.87	0.9061
FPN	88.13	98.38	96.00	93.69	0.9114
PSPNet	80.49	97.22	94.41	87.14	0.8308
PAN	85.20	97.97	96.04	90.62	0.8777
DeepLabV3+	89.63	98.50	96.20	93.50	0.9103

**Table 7 plants-13-03186-t007:** Training time for the selected deep learning models.

Model	Time/min
U-Net	122.75
U-Net++	100.27
Manet	171.54
Linknet	86.3
FPN	80.66
PSPNet	96.81
PAN	86.1
DeepLabV3+	105.45

## Data Availability

The datasets generated during and/or analyzed during the current study are available from the corresponding author upon reasonable request.

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
