# Peer review of "Accurately Segmenting/Mapping Tobacco Seedlings Using UAV RGB Images Collected from Different Geomorphic Zones and Different Semantic Segmentation Models"

_plants, 2024, doi:10.3390/plants13223186_

Round 1
Reviewer 1 Report
Comments and Suggestions for Authors
This paper is interesting and applied 8 eight models for semantic segmentation of tobacco plants from UAV images. The experiment is very interesting, but this paper draft has a lot of room for improvement in writing and organization. The comments and questions are summarized as follows:
1) In Abstract, line 15-16, the description can be improved, it maybe “this study explores an accurate segmentation model for detecting tobacco seedlings from UAV RGB images across various geomorphic partitions, including dam and hilly areas.”
2) Line 18, “the DASTD (Hillside Seedling Tobacco Dataset)” it should be wrong here.
3) Line 32, the author may use three digits of accuracy.
4) Line 47-48, the description of “1 million hectares” may be repeated.
5) Line 174-175, the font of °, ′ and ″ maybe not consistent.
6) Line 194-195, “can be equipped” may be “The DJI Mavic 2 Pro drone is compact and ultra-lightweight, capable of being equipped with a variety of sensors, including the 1-inch CMOS Hasselblad camera. This camera is ideal for capturing high-quality remote sensing data, such as visible light (RGB) imagery, which can be utilized for monitoring tobacco plants in the study area.”
(7) Line 211, “handbook” maybe wrong here.
(8) Line 212, give a full name of DOM and DSM.
(9) Line 217-218 , “directly” was used two times here.
(10) Line 224, “into” was repeated.
(11) Line 310, “to segmentation of” may be “to the segmentation of “ or “to segment seedling tobacco plants”.
(12) The data analysis of 3.1 and 3.2 is too complicated and can be made simpler. The author can simply and summarize key points of the data comparison.
Comments on the Quality of English LanguageI believe the language in this article could be more coherent and concise. There are several minor mistakes and grammatical errors.
Author Response
Authors' Response to Reviewers' Comments
Revision notes
Here are our detailed responses to the reviews. Please note that the comments from the reviewers are in BLACK followed by our responses in BLUE text.
We would like to thank you cordially for your thorough and constructive comments concerning our manuscript entitled “Accurately Segmenting/Mapping Tobacco Seedlings Using UAV RGB Images Collected from Different Geomorphic Zones and Different Semantic Segmentation Models” (Plants-3244388). We do feel that those comments are all valuable and helpful for us to improve this manuscript. With the guidance of the comments, we made a quite extensive revising in this revision with details as following:
Reviewer #1:
This paper is interesting and applied 8 eight models for semantic segmentation of tobacco plants from UAV images. The experiment is very interesting, but this paper draft has a lot of room for improvement in writing and organization. The comments and questions are summarized as follows:
Response: We are extremely grateful for your positive comments and valuable criticisms.
Comments 1: In Abstract, line 15-16, the description can be improved, it maybe “this study explores an accurate segmentation model for detecting tobacco seedlings from UAV RGB images across various geomorphic partitions, including dam and hilly areas.”
Response 1:Thank you for pointing this out. We agree with this comment. We have checked the abstract and changed the expression to “this study explores an accurate segmentation model for detecting tobacco seedlings from UAV RGB images across various geomorphic partitions, including dam and hilly areas.”. Corresponding to Line 15-17.
Comments 2: Line 18, “the DASTD (Hillside Seedling Tobacco Dataset)” it should be wrong here.
Response 2: Thank you for pointing this out. We have checked “the DASTD (Hillside Seedling Tobacco Dataset)”and changed the expression to“the DASTD (Dam Area Seedling Tobacco Dataset)”. Corresponding to Line 19.
Comments 3: Line 32, the author may use three digits of accuracy.
Response 3: Thank you for pointing this out. We have used two digits of accuracy, that is “H-DASTD (0.901) > DASTD (0.885) > HSTD (0.726).” , Simultaneously apply to the entire text. Corresponding to Line 33.
Comments 4: Line 47-48, the description of “1 million hectares” may be repeated.
Response 4: Thank you for pointing this out. We have removed duplicate content. Corresponding to Line 46-47.
Comments 5: Line 174-175, the font of °, ′ and ″ maybe not consistent.
Response 5: Thank you for pointing this out. We have changed the format to be consistent. Amend “(25.18°N, 105.19°E)" to "(25°10'48''N, 105°11'24''E)”. Corresponding to Line 173.
Comments 6: Line 194-195, “can be equipped” may be “The DJI Mavic 2 Pro drone is compact and ultra-lightweight, capable of being equipped with a variety of sensors, including the 1-inch CMOS Hasselblad camera. This camera is ideal for capturing high-quality remote sensing data, such as visible light (RGB) imagery, which can be utilized for monitoring tobacco plants in the study area.”
Response 6: Thank you for pointing this out. We have revised the content in the main text to “The DJI Mavic 2 Pro drone is compact and ultra-lightweight, capable of being equipped with a variety of sensors, including the 1-inch CMOS Hasselblad camera. This camera is ideal for capturing high-quality remote sensing data, such as visible light (RGB) imagery, which can be utilized for monitoring tobacco plants in the study area.”. Corresponding to Line 192-195.
Comments 7: Line 211, “handbook” maybe wrong here.
Response 7: Thank you for pointing this out. We have changed it to “Smart Handbook”. Corresponding to Line 210.
Comments 8: Line 212, give a full name of DOM and DSM.
Response 8: Thank you for pointing this out. We have modified “DOM and DSM” to “Digital Orthophoto Map (DOM) and (b) Digital Surface Model (DSM).”. Corresponding to Line 211-212.
Comments 9: Line 217-218 , “directly” was used two times here.
Response 9: Thank you for pointing this out. We have removed the second “directly”.
Corresponding to Line 218.
Comments 10: Line 224, “into” was repeated.
Response 10: Thank you for pointing this out. We have removed the duplicate“into”. Corresponding to Line 223.
Comments 11: Line 310, “to segmentation of” may be “to the segmentation of “ or “to segment seedling tobacco plants”.
Response 11: Thank you for pointing this out. We have changed “to segmentation of” to “to the segmentation of”. Corresponding to Line 297-298.
Comments 12: The data analysis of 3.1 and 3.2 is too complicated and can be made simpler. The author can simply and summarize key points of the data comparison.
Response 12: Thank you for pointing this out. We have carefully considered your feedback and summarized the key points and simplified sections of 3.1 and 3.2. However, due to the use of 8 semantic segmentation models in this study to accurately segment tobacco seedlings in dam and hilly areas. Therefore, the analysis of the results involves a lot of content. If too much content is simplified, it may lead to incomplete expression of the research content. Therefore, we conducted a simple optimization.
Comments 13: I believe the language in this article could be more coherent and concise. There are several minor mistakes and grammatical errors.
Response 13: Thank you for pointing this out. The article has been checked and revised by English experts in the relevant fields.

Reviewer 2 Report
Comments and Suggestions for Authors
The paper "Accurate Segmentation of Tobacco Seedlings Based on Semantic Segmentation Model and UAV RGB Images under Different Geomorphic Zones’’ explores an accurate segmentation model for tobacco seedlings based on UAV RGB images based on geomorphic partitions (dam and hilly areas). Exploring the family of tobacco plant seedling segmentation networks: U-Net, U-Net++, Linknet, PSPNet, MAnet, FPN, PAN, and DeepLabV3+. Based on the HSTD (Hillside Seedling Tobacco Dataset), the DASTD (Hillside Seedling Tobacco 19 Dataset), and the Hillside-Damside Seedling Tobacco Dataset (H-DASTD) for model training. In order to validate the performance of the semantic segmentation model for crop segmentation in complex cropping environments in Karst mountainous areas, the study compares and analyses the predicted results with the manually labeled true value. The study shows that accurate segmentation of tobacco plant seedlings in dam and hilly areas based on UAV RGB images and semantic segmentation model provides a new idea and technical support for accurate crop segmentation in Karst mountainous.
The paper is well-organized and clearly explains the proposed method and its theoretical underpinnings. The authors thoroughly review related work in the field and present experimental results that demonstrate the effectiveness of their approach. Comparing existing methods strengthens the paper's conclusions.
The paper is a valuable contribution. Its novel approach and strong experimental results make it a worthwhile read for researchers and practitioners in the field.
There are the following comments hopefully to be adopted:
- Discuss the implications of the proposed method and how they contribute to the existing literature.
- A discussion on any work limitations is missing and should be included in the final version of the paper.
Based on the above, it is recommended that the authors review the entire document, considering the suggestions mentioned above as well as any others that other reviewers may propose.
All the best.
Author Response
Authors' Response to Reviewers' Comments
Revision notes
Here are our detailed responses to the reviews. Please note that the comments from the reviewers are in BLACK followed by our responses in BLUE text.
We would like to thank you cordially for your thorough and constructive comments concerning our manuscript entitled “Accurately Segmenting/Mapping Tobacco Seedlings Using UAV RGB Images Collected from Different Geomorphic Zones and Different Semantic Segmentation Models” (Plants-3244388). We do feel that those comments are all valuable and helpful for us to improve this manuscript. With the guidance of the comments, we made a quite extensive revising in this revision with details as following:
Reviewer#2:
The paper "Accurate Segmentation of Tobacco Seedlings Based on Semantic Segmentation Model and UAV RGB Images under Different Geomorphic Zones’’ explores an accurate segmentation model for tobacco seedlings based on UAV RGB images based on geomorphic partitions (dam and hilly areas). Exploring the family of tobacco plant seedling segmentation networks: U-Net, U-Net++, Linknet, PSPNet, MAnet, FPN, PAN, and DeepLabV3+. Based on the HSTD (Hillside Seedling Tobacco Dataset), the DASTD (Hillside Seedling Tobacco 19 Dataset), and the Hillside-Damside Seedling Tobacco Dataset (H-DASTD) for model training. In order to validate the performance of the semantic segmentation model for crop segmentation in complex cropping environments in Karst mountainous areas, the study compares and analyses the predicted results with the manually labeled true value. The study shows that accurate segmentation of tobacco plant seedlings in dam and hilly areas based on UAV RGB images and semantic segmentation model provides a new idea and technical support for accurate crop segmentation in Karst mountainous.
The paper is well-organized and clearly explains the proposed method and its theoretical underpinnings. The authors thoroughly review related work in the field and present experimental results that demonstrate the effectiveness of their approach. Comparing existing methods strengthens the paper's conclusions.
The paper is a valuable contribution. Its novel approach and strong experimental results make it a worthwhile read for researchers and practitioners in the field.
There are the following comments hopefully to be adopted:
Response: We are extremely grateful for your positive comments and valuable criticisms.
Comments 1: Discuss the implications of the proposed method and how they contribute to the existing literature.
Response 1: Thank you for pointing this out.
We have added a relevant review on tobacco segmentation methods in our discussion, comparing and analyzing the research results with those of other authors. Added a section on the research content titled “4.4 Differences from existing research”. Corresponding to Line 651-681.
The results of this study indicate that semantic segmentation models can effectively segment tobacco seedlings when trained on low-altitude drone images. Considering the challenging conditions brought by the topography and landforms of Karst mountainous areas (e.g., dam areas and hilly areas), this study found that semantic segmentation models performed better in segmenting tobacco seedlings in the dam hilly area than in the hilly area. It is difficult to consistently segment tobacco seedlings in the hilly area because of the complex plots, significant differences in the growth of tobacco seedlings, and diversity of tobacco seedling plants in the labeled images. Compared with tobacco seedlings in the dam area, the tobacco contours are less distinct and there are more weeds. Although including the annotation of a large number of tobacco seedling samples when training models can help increase sample diversity and improve segmentation accuracy, terrain fragmentation in hilly areas still has an impact on model segmentation of tobacco seedlings.
Although there are few studies that show direct similarities with this study, we comment on other works in the relevant literature and how they compare to ours. Liang Huang et al. [8] used DeeplabV3+, PSPNet, SegNet, and U-Net to segment tobacco planting areas in high-altitude mountainous areas, and obtained mIoU values of 0.9436, 0.9118, 0.9392, and 0.9473, respectively, verifying the feasibility of using semantic segmentation models to extract tobacco planting areas from drone remote sensing images. Xiaodong Bai et al. [42] designed an RPNet model to count rice plants after the tillering stage and compared this model with MCNN, CSRNet, SANet, TasselNetV2, and FIDTM. The MAE, RMSE, rMAE, and rRMSE increased by 8.3%, 11.2%, 1.2%, and 1.6%, respectively. Therefore, RPNet can be used to accurately count rice plants in paddy fields, replacing traditional manual counting methods. Jie Li et al. [43] designed RapeNet and RapeNet+ models for automatic counting of rapeseed flower clusters. The experimental results showed that the RapeNet series outperformed other state-of-the-art counting methods, providing important technical support for field rapeseed flower cluster counting statistics. In existing research, Liang Huan et al.'s method is the most similar to ours; although their method segments different objects, it uses the same model for segmentation and achieves better segmentation accuracy.
Comments 2: A discussion on any work limitations is missing and should be included in the final version of the paper.
Response 2: Thank you for pointing this out. We have added a section on “4.5 Limitations of the Study” in the discussion, which outlines the limitations of this study. Corresponding to Line 682-705.
4.5. Limitations of the Study
This study used semantic segmentation models to segment tobacco seedlings in dam and hilly areas. Although good results were achieved, there are still limitations, mainly in the following aspects:
(1) The terrain in Karst mountainous areas is fragmented, and the composition of near-ground objects is complex, with staggered power lines, signal base stations, and other objects. It is difficult for drones to collect high-resolution images by flying at ultra-low altitudes close to the ground. Due to interference factors such as meteorology, topography, and signal base stations that affect the safe flight of drones, the spatial resolution of images may not be able to meet the precise segmentation of tobacco seedlings. For practical applications, factors such as the spatial resolution of data, the planting structure of the research area, and fragmentation of farmland should be considered to determine the optimal monitoring time and required image spatial resolution, as well as to develop the corresponding segmentation method with the lowest cost to meet the accuracy requirements for segmenting tobacco seedlings.
(2) The results of this study are based only on drone images from two areas: a dam area and a hilly area. Although this study selected tobacco seedling planting sites based on the topography and landforms of Karst mountainous areas, it cannot guarantee that these sites fully represent other tobacco planting areas in Karst mountainous areas. Similarly, obtaining high-resolution drone images is difficult, and a large number of tobacco seedling plant samples cannot be obtained. Although various processing methods were applied to the samples to increase their diversity, they cannot represent all types of tobacco cultivation in Karst mountainous areas, which affects model training and reduces the performance of model segmentation.
Comments 3: Based on the above, it is recommended that the authors review the entire document, considering the suggestions mentioned above as well as any others that other reviewers may propose.
Response 3: Thank you for pointing this out. We have made modifications and added relevant content based on your suggestions.

Reviewer 3 Report
Comments and Suggestions for Authors
The authors sought to accurately segment tobacco seedlings using several semantic segmentation models on UAV RGB images collected from different geomorphic zones. The method presented is sound and I can recommend publication pending a few corrections. My comments are given below.
The title could be changed or improved. Consider using the following or some variant of it: Accurately segmenting/mapping tobacco seedlings using UAV RGB images collected from different geomorphic zones and different semantic segmentation models.
Several sentences presented in the manuscript were very long and should be reduced. Also, in some instances the writing was awkward and difficult to understand. Table 2 could be split into two or three figures that can fit on a single page.
Also, consider dropping the Kappa results as it has been long established that Kappa is not a good measure of classification accuracy. Please see: Pontius, R.G. and Millones, M. (2011) Death to Kappa: Birth of Quantity Disagreement and Allocation Disagreement for Accuracy Assessment. International Journal of Remote Sensing, 32, 4407-4429.
The discussion has too much information that should/could be included as results. The authors should use the discussion to put their results in context. I.e., how does their results compare with other studies that have tried to do the same using similar or different methods on tobacco seedlings (did the other studies that used similar or different methods get similar accuracies or encounter similar problems with hilly vs flat areas?), or even use the same deep learners on RGB images when mapping other features. There were no cited studies included in the discussion, which is highly unusual.
Please state the program/software and the version that was used to do the segmentation. Can I assume it was Python?

Several sentences presented in the manuscript were very long and should be reduced. Also, in some instances the writing was awkward and difficult to understand. I have included a pdf with some suggested editing (they were included as comments).
Author Response
Authors' Response to Reviewers' Comments
Revision notes
Here are our detailed responses to the reviews. Please note that the comments from the reviewers are in BLACK followed by our responses in BLUE text.
We would like to thank you cordially for your thorough and constructive comments concerning our manuscript entitled “Accurately Segmenting/Mapping Tobacco Seedlings Using UAV RGB Images Collected from Different Geomorphic Zones and Different Semantic Segmentation Models” (Plants-3244388). We do feel that those comments are all valuable and helpful for us to improve this manuscript. With the guidance of the comments, we made a quite extensive revising in this revision with details as following:
Reviewer#3 :
The authors sought to accurately segment tobacco seedlings using several semantic segmentation models on UAV RGB images collected from different geomorphic zones. The method presented is sound and I can recommend publication pending a few corrections. My comments are given below.
Response: We are extremely grateful for your positive comments and valuable criticisms.
Comments 1: The title could be changed or improved. Consider using the following or some variant of it: Accurately segmenting/mapping tobacco seedlings using UAV RGB images collected from different geomorphic zones and different semantic segmentation models.
Response 1: Thank you for pointing this out. We agree with this comment. We have changed the title to “Accurately Segmenting/Mapping Tobacco Seedlings Using UAV RGB Images Collected from Different Geomorphic Zones and Different Semantic Segmentation Models.” ,which we believe will be more corresponding to the content of the paper. Corresponding to Line 2-4.
Comments 2: Several sentences presented in the manuscript were very long and should be reduced. Also, in some instances the writing was awkward and difficult to understand. Table 2 could be split into two or three figures that can fit on a single page.
Response 2: Thank you for pointing this out. We have carefully reviewed the entire manuscript and made revisions to long sentences. Due to the large size of the figures in Table 2, which contain a lot of information, if the figures in Table 2 are adjusted to one page, problems such as unclear information in the figures may occur. Therefore, the figures were not adjusted to the same page.
Comments 3: Also, consider dropping the Kappa results as it has been long established that Kappa is not a good measure of classification accuracy. Please see: Pontius, R.G. and Millones, M. (2011) Death to Kappa: Birth of Quantity Disagreement and Allocation Disagreement for Accuracy Assessment. International Journal of Remote Sensing, 32, 4407-4429.
Response 3: Thank you for pointing this out. We use the Kappa coefficient as one of the evaluation indicators in the manuscript, which helps to compare and analyze the segmentation accuracy of the model with several other indicators. Although it is not the optimal evaluation indicator, it can analyze the segmentation accuracy of the model on tobacco seedlings. Therefore, this manuscript retains the Kappa coefficient as an evaluation indicator.
Comments 4: The discussion has too much information that should/could be included as results. The authors should use the discussion to put their results in context. I.e., how does their results compare with other studies that have tried to do the same using similar or different methods on tobacco seedlings (did the other studies that used similar or different methods get similar accuracies or encounter similar problems with hilly vs flat areas?), or even use the same deep learners on RGB images when mapping other features. There were no cited studies included in the discussion, which is highly unusual.
Response 4: Thank you for pointing this out.
(1) We have summarized the content of the discussion and included it in the results. The added content is: “(3) Through an analysis of the performance of U-Net, U-Net++, Linknet, PSPNet, MAnet, FPN, PAN, and DeepLabV3+ in tobacco segmentation in the dam and hilly areas, it was found that factors such as topography and land fragmentation have significant impacts on model errors and omissions during segmentation. U-Net, U-Net++, Linknet, MAnet, and FPN showed better segmentation performance, with missed segmentation of 2867, 2986, 2933, 3164, and 2813 plants in the hilly area and 149, 135, 134, 147, and 194 plants in the dam area, respectively. This means that U-Net, U-Net++, Linknet, MAnet, and FPN are more suitable for the segmentation of crop seedlings in dam areas. The misclassification of tobacco seedlings in the hilly area was much higher than in the dam area, with PSPNet showing the smallest difference at 341 plants, while PAN showed the largest difference at 6568 plants. Among the eight segmentation models, Linknet and PSPNet had similar numbers of misclassifications in the dam and hilly areas, and the terrain, landforms, and fragmentation of land parcels had little impact on them. U-Net, U-Net++, MAnet, FPN, PAN, and DeepLabV3+ were greatly affected by factors such as terrain, landforms, and land fragmentation. Therefore, for Karst mountainous areas with diverse landforms, accurate crop segmentation is mainly influenced by terrain and landforms, the effects of which cannot be generalized and require zoning research.”. Corresponding to Line 737-753.
(2)We have added a relevant review on tobacco segmentation methods in our discussion, comparing and analyzing the research results with those of other authors. Added a section on the research content titled “4.4 Differences from existing research”.Corresponding to Line 651-681.
4.4. Differences from existing research
The results of this study indicate that semantic segmentation models can effectively segment tobacco seedlings when trained on low-altitude drone images. Considering the challenging conditions brought by the topography and landforms of Karst mountainous areas (e.g., dam areas and hilly areas), this study found that semantic segmentation models performed better in segmenting tobacco seedlings in the dam hilly area than in the hilly area. It is difficult to consistently segment tobacco seedlings in the hilly area because of the complex plots, significant differences in the growth of tobacco seedlings, and diversity of tobacco seedling plants in the labeled images. Compared with tobacco seedlings in the dam area, the tobacco contours are less distinct and there are more weeds. Although including the annotation of a large number of tobacco seedling samples when training models can help increase sample diversity and improve segmentation accuracy, terrain fragmentation in hilly areas still has an impact on model segmentation of tobacco seedlings.
Although there are few studies that show direct similarities with this study, we comment on other works in the relevant literature and how they compare to ours. Liang Huang et al. [8] used DeeplabV3+, PSPNet, SegNet, and U-Net to segment tobacco planting areas in high-altitude mountainous areas, and obtained mIoU values of 0.9436, 0.9118, 0.9392, and 0.9473, respectively, verifying the feasibility of using semantic segmentation models to extract tobacco planting areas from drone remote sensing images. Xiaodong Bai et al. [42] designed an RPNet model to count rice plants after the tillering stage and compared this model with MCNN, CSRNet, SANet, TasselNetV2, and FIDTM. The MAE, RMSE, rMAE, and rRMSE increased by 8.3%, 11.2%, 1.2%, and 1.6%, respectively. Therefore, RPNet can be used to accurately count rice plants in paddy fields, replacing traditional manual counting methods. Jie Li et al. [43] designed RapeNet and RapeNet+ models for automatic counting of rapeseed flower clusters. The experimental results showed that the RapeNet series outperformed other state-of-the-art counting methods, providing important technical support for field rapeseed flower cluster counting statistics. In existing research, Liang Huan et al.'s method is the most similar to ours; although their method segments different objects, it uses the same model for segmentation and achieves better segmentation accuracy.
Comments 5: Please state the program/software and the version that was used to do the segmentation. Can I assume it was Python?
Response 5: Thank you for pointing this out. We use Python language for model building, meanwhile, We have added a section on “2.5. Training environment”. Corresponding to Line 252-260.
2.5. Training environment
During the training process, a semantic segmentation model performs numerous calculations and occupies memory and video memory features; thus, the operation of the model poses high hardware requirements. In the study framework, we used TensorFlow 2.0.0 to build a DL research environment and test the U-Net, U-Net++, Linknet, PSPNet, MAnet, FPN, PAN, and DeepLabV3+ models. The hardware and software environments are described in Table 3.
Table 3. Details on hardware and software environments.
|
Basic configuration project |
Basic configuration content |
Software configuration project |
Software version |
|
System |
Windows 10 professional edition |
GPU-Drive |
560.94 |
|
CPU |
13th Gen Intel(R) Core(TM) i7-13700KF 3.40 GHz |
CUDA |
12.6 |
|
RAM |
32GB |
Python |
3.7 |
|
Hard disk |
1940GB |
Tensorflow |
2.0.0 |
|
Graphics card |
NVIDIA GeForce RTX4080 |
Torch |
1.7.1 |
Comments 6: Several sentences presented in the manuscript were very long and should be reduced. Also, in some instances the writing was awkward and difficult to understand. I have included a pdf with some suggested editing (they were included as comments).
Response 6: Thank you for pointing this out. We have made modifications to the lengthy sentences in the manuscript. Meanwhile, the article has been checked and revised by English experts in the relevant fields. The manuscript content has been revised as follows:
Comments (1): Please rephrase. Difficult to understand. “Tobacco seedling is the key period of tobacco planting, accurate extraction of tobacco seedling can effectively help farmers replanting, accurate fertilization and help in the later stage of yield estimation.”
Response (1):We have modified it to “The tobacco seedling stage is a crucial period for tobacco cultivation. Accurately extracting tobacco seedlings from satellite images can effectively assist farmers in replanting, precise fertilization, and subsequent yield estimation.”. Corresponding to Line 10-12.
Comments (2): Sentence is too long. Consider splitting in twoor shortening it.
1): As an important economic crop, it has been planted in more than 125 countries around the world, with an estimated planting area of more than 4 million hectares, and as the largest country of tobacco cultivation in China, the tobacco planting area involves more than 1700 counties and municipalities in 27 provinces, and the area of tobacco planting in 2021 reaches nearly 1 million hectares.
Response 1):We have modified it to “As an important economic crop, it is planted in over 125 countries worldwide, with a global estimated planting area of over 4 million hectares. China is the largest country in tobacco cultivation, with areas in over 1700 counties and cities in 27 provinces being used for this purpose.” Corresponding to Line 44-47.
2): Affected by the terrain undulation, surface fragmentation, gully and soil-forming factors, the territory of the sloping arable land, large area, wide distribution and dispersion; at the same time, the Karst mountainous areas are crowded and few, mountainous and flat, most of the tobacco planted in the terrain is complex, the land parcel is broken on the sloping arable land, the planting space distribution is scattered, the background characteristics of the land parcels are complex, and the crop mixing is extremely high; at the same time, the risk of the process of tobacco planting is high, and it is susceptible to At the same time, the tobacco planting process is risky and vulnerable to natural disasters and pests.
Response 2):We have modified it to “Because of its undulating terrain, fragmented surface, vertical and horizontal valleys, and soil-forming factors, there are many large sloping farmland areas with a wide and scattered distribution within the territory. At the same time, the Karst mountainous areas have more people and less flat land. Tobacco is mostly planted on sloping farmland with a complex terrain and fragmented plots. The distribution of planting space is scattered, the background characteristics of planting plots are complex, and there are numerous cases of crop intercropping. Furthermore, the tobacco cultivation process carries high risks, as it is susceptible to natural disasters and pests.” Corresponding to Line 61-69.
3): And the tobacco seedling period can timely check the growth of tobacco seedlings after transplanting, for dead seedlings, weak seedlings (pulled out in time) or lack of seedlings, etc. should be timely replanting; for the fading of tobacco seedlings can be timely watering in an appropriate amount, in order to facilitate the normal growth of seedlings [12, 13].
Response 3): We have modified it to “During the tobacco seedling stage, the growth of transplanted tobacco seedlings can be checked in a timely manner. Timely replanting should be carried out for dead, weak, or missing seedlings. It is advisable to water withered tobacco seedlings in a timely manner to facilitate their normal growth [12, 13]” Corresponding to Line 81-84.
4): Because UAVs can carry multispectral, hyperspectral, visible sensors, etc., and are not subject to the limitations of space, time, spectral resolution, and weather complexity, they can acquire high spatial resolution images of the complex surface in Karst mountainous areas in a flexible, economical, and efficient way, making up for the short board of satellite images that cannot obtain real-time high-resolution images, and providing spatial distribution information of crops for precision agriculture, and thus remote sensing from UAVs has been Therefore, UAV remote sensing is widely used in precision agriculture [18, 19, 20, 21, 22, 23].
Response 4):We have modified it to “The main reason is the ability of drones to be equipped with sensors that capture images across multispectral, hyperspectral, and visible light, while being unrestricted by spatial, temporal, and spectral resolutions, as well as complex weather. Therefore, high-spatial resolution images of complex terrain in karst mountainous areas can be obtained in a flexible, economical, and efficient manner, compensating for the shortcoming of not being able to obtain real-time high-resolution satellite images. As UAV remote sensing provides spatial distribution information of crops, it has been widely used in precision agriculture [18, 19, 20, 21, 22, 23].” Corresponding to Line 111-118.
5): However, when counting tobacco seedlings, the number of seedlings in a piece of land can be as few as a few, or as many as thousands, especially in the Karst mountainous areas with fragmented plots and complex surfaces, where tobacco is planted in environments with many weeds and mixed crops, it is difficult to accurately segment tobacco seedlings, and in addition, the large number of tobacco seedling samples has brought a huge amount of workload to the labeling work.
Response 5):We have modified it to “However, the number of tobacco seedlings in a field can range from a few to thousands, especially in Karst mountainous areas where the land is fragmented and the surface is complex. Tobacco is planted in environments with numerous weeds and mixed crops, making it difficult to accurately segment tobacco seedlings. In addition, the large number of tobacco seedling samples brings enormous workload to the labeling process.” .Corresponding to Line 151-156.
Comments (3): It looks like over all PSPNet did not perform better than DeepLabV3+.Consider swapping them around.
a difference of 0.213. Therefore, the results of the study show that U-net, U-Net++, Linknet, and PSPNet have better overall performance than MAnet, FPN, PAN, and DeepLabV3+.
Response (3):We have modified it to “Therefore, the research results indicate that the overall segmentation performance of U-net, U-Net++, Linknet, and PSPNet is better than that of MAnet, FPN, PAN, and DeepLabV3+.” Corresponding to Line334-336.
Comments (4): What is the purpose of this line? ls this a subheading or a listing? Please rephrase and reformat.
Leakage of split tobacco seedlings (Figure 10(a)), the leakage of split tobacco seedlings in the dam area is much less than the leakage of split tobacco bead seedlings in the hilly area.
Response(4):Thank you for pointing this out. The purpose of this sentence is to summarize and outline the content of the entire paragraph, followed by an explanation of the sentence. It is neither a subtitle nor a list. We have modified it to “The analysis of missed segmentation of tobacco seedlings revealed (Figure 10 (a)) that the number of tobacco seedlings that were missed in the dam area was much less than the number of tobacco seedlings that were missed in the hilly area.” Corresponding to Line 526-528.
Comments (5): Is this a subheading or a listing? Please rephrase and reformat.
Mis-segmentation of tobacco seedlings (Figure 10(b)), tobacco seedlings mis-segmentation hilly area is much higher than the dam area, PSPNet has the smallest difference of 341 plants, and PAN has the largest difference of 6,568 plants.
Response (5): Thank you for pointing this out. This sentence is neither a subtitle nor a list, but a summary and overview of this paragraph. We have modified it to “The analysis of misclassified tobacco seedlings revealed (Figure 10 (b)) that mis-segmentation of tobacco seedlings in the hilly area was much higher than in the dam area. PSPNet showed the smallest difference of 341 plants, while PAN showed the largest difference of 6,568 plants.” Corresponding to Line 549-552.

Round 2
Reviewer 1 Report
Comments and Suggestions for Authors
The author has made a lot of changes, it looks good.